# On-tissue dataset-dependent MALDI-TIMS-MS² bioimaging

Steffen Heuckeroth [1], Arne Behrens [2], Carina Wolf[1], Arne Fütterer [2], Ilona D. Nordhorn [1], Katharina Kronenberg [1], Corinna Brungs[3], Ansgar Korf[2], Henning Richter [4], Astrid Jeibmann[5], Uwe Karst [1] & Robin Schmid [3,6] ✉

Trapped ion mobility spectrometry (TIMS) adds an additional separation dimension to mass spectrometry (MS) imaging, however, the lack of fragmentation spectra (MS²) impedes confident compound annotation in spatial metabolomics. Here, we describe spatial ion mobility-scheduled exhaustive fragmentation (SIMSEF), a dataset-dependent acquisition strategy that augments TIMS-MS imaging datasets with MS² spectra. The fragmentation experiments are systematically distributed across the sample and scheduled for multiple collision energies per precursor ion. Extendable data processing and evaluation workflows are implemented into the open source software MZmine. The workflow and annotation capabilities are demonstrated on rat brain tissue thin sections, measured by matrix-assisted laser desorption/ionisation (MALDI)-TIMS-MS, where SIMSEF enables on-tissue compound annotation through spectral library matching and rule-based lipid annotation within MZmine and maps the (un)known chemical space by molecular networking. The SIMSEF algorithm and data analysis pipelines are open source and modular to provide a community resource.

Mass spectrometry (MS) imaging gains traction in metabolic, lipidomic, and proteomic studies[1] since the analyte distribution in biological tissues grants valuable insights into how a disease affects an organism. In recent years, advances in instrumentation led to higher spatial resolution[2], increased specificity[3], boosted sensitivity[4], and increased sample throughput[5,6]. These characteristics are linked and influence each other, for example, higher spatial resolutions require efficient ionisation and increased sensitivity. A major bottleneck in many MS imaging studies remains the compound annotation, which often relies on accurate mass only, due to missing fragmentation data (MS²)[2,7,8]. Therefore, spectral library matching, manual annotation of fragmentation spectra, molecular networking[9,10], and other fragment ion-based approaches, such as molecular structure prediction in the SIRIUS software, remain unavailable for these MS¹-only workflows[11–13]. Generally, most public

MS imaging studies lack MS² data because of the limited availability of spatially resolved data-dependent acquisition (DDA) modes, where MS² scans are usually scheduled based on the most abundant signals detected in MS¹. While some studies have investigated MS imaging experiments with DDA, they rely on DDA methods initially designed for liquid chromatography (LC)-MS. Thus, they cannot leverage the spatial distribution for ideal MS² coverage[14–17]. Nevertheless, this approach allowed more confident metabolite annotation based on low-resolution MS² spectra acquired together with high-resolution MS¹ images[18]. Other studies use a targeted approach with predefined isolation windows to acquire MS²-only images[19], or manually set up inclusion lists and cycle MS¹ and targeted MS² experiments[15,16,20]. These strategies generate fragment ion information but are limited to a few targets leaving the chemical composition underexplored.

[1]Institute of Inorganic and Analytical Chemistry, University of Münster, Münster, Germany. [2]Bruker Daltonics GmbH & Co. KG, Bremen, Germany. [3]Institute of Organic Chemistry and Biochemistry of the Czech Academy of Sciences, Prague, Czech Republic. [4]Clinic for Diagnostic Imaging, Diagnostic Imaging Research Unit (DIRU), University of Zurich, Zürich, Switzerland. [5]Institute of Neuropathology, University Hospital Münster, Münster, Germany. [6]Collaborative Mass Spectrometry Innovation Center, University of California San Diego, La Jolla, CA, USA. ✉e-mail: robin.schmid@uochb.cas.cz

Recently, ion mobility spectrometry (IMS)-MS is gaining popularity in proteome and metabolome research[21–24]. The additional separation dimension based on the collision cross section (CCS)-to-charge ratio improves compound annotation and enables additional data acquisition strategies. A promising IMS technology is trapped IMS (TIMS), which traps ions in an electrostatic field and a counter gas flow[25,26]. The TIMS analyser consists of two separated IMS regions, one to accumulate ions and the other to separate and release an accumulated ion package. This technique has been incorporated into quadrupole time-of-flight (qTOF)-MS instruments with fast precursor isolation switching times. Synchronising the quadrupole isolation with the mobility separation enabled the development of additional DDA modes, such as parallel accumulation serial fragmentation (PASEF)[27,28]. Several additional PASEF modes have been described recently, such as data-independent acquisition (dia)PASEF[29], parallel reaction monitoring (prm)PASEF[30], midiaPASEF[31], slicePASEF[32], and synchroPASEF[33]. While these techniques provide improvements in the field of proteomics, they are designed and exclusively available in LC-IMS-MS analysis. Recently, use cases of a prototypic prm-MALDI workflow have been described to acquire multiple fragmentation spectra in a single TIMS ramp in a targeted analysis[34]. Thereby, fragment ion images of multiple precursor ions were acquired simultaneously. However, this approach relied on prior, manual data evaluation, precursor selection, hand-written precursor schedules, and manual data acquisition.

Comprehensive spatial metabolomics studies acquire complementary datasets by LC-MS[2] and MS[1] imaging, without fragmentation analysis. Here, laser capture microdissection enables sampling of regions from tissue thin sections for subsequent extraction and LC-MS analysis[35–37]. Integrative data analysis then aligns LC-MS[2] features with ion images by their *m/z*. Furthermore, ion mobility or CCS can act as additional identifiers to align LC-IMS-MS and MALDI-IMS-MS imaging experiments by more than just *m/z*[38]. MZmine 3 is, to the best of our knowledge, the first tool to provide such an integrative data analysis workflow that results in a combined feature list with MS[2] from LC-IMS-MS and spatial distributions from MS imaging[39]. However, this approach requires extensive sample preparation, which must be adequate to extract the metabolites of interest. Furthermore, the observed ion species (adduct) in MALDI-MS often differs from the adduct observed after electrospray ionisation, altering *m/z* and CCS values, effectively hampering a direct alignment.

Addressing the need for dataset-dependent MS[2] acquisition in TIMS-MS imaging, we introduce workflows for spatial ion mobility-scheduled exhaustive fragmentation (SIMSEF) analysis. SIMSEF is wrapped into open source MZmine 3 modules to plan, acquire, and analyse spatially-resolved MS[2] data. First, MALDI-TIMS-qTOF-MS analysis produces TIMS-MS[1] images, which are analysed for subsequent, automated MS[2] experiments to achieve an exhaustive precursor coverage. By tracking spatial coordinates, the module was designed to optimise the scheduling of MS[2] events across a sample, considering the expected precursor purity of an isolation window, mobility separation, and spatial distribution. The final MS[2]-enriched datasets can be visualised and evaluated in MZmine 3, offering compound annotation and data integration with other platforms such as SIRIUS and Global Natural Products Social Molecular Networking (GNPS). The GNPS web platform enables molecular networking and integration of the results into a broader MS ecosystem. We anticipate the SIMSEF workflow will enrich (public) MS imaging data by providing confident in-depth compound annotation by high-quality MS[2] spectra paired with CCS values. This study is to stimulate developments in dataset-dependent acquisition strategies for MS imaging by delivering open source solutions.

## Results

A non-target workflow for the dataset-dependent acquisition of MS[2] spectra for MALDI-TIMS-MS imaging experiments was developed. The workflow is illustrated in Fig. 1. After TIMS-MS[1] imaging data acquisition, the non-targeted feature detection in MZmine 3 creates a feature

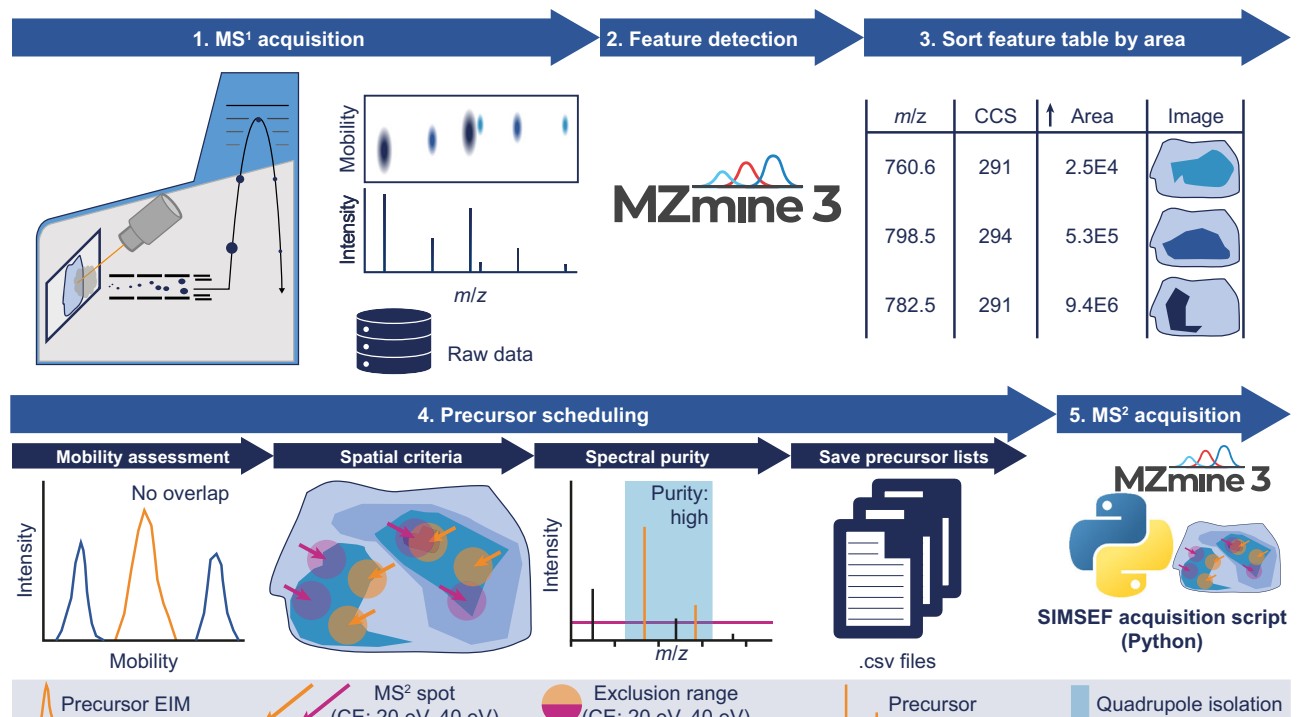

**Fig. 1 | Schematic overview of the MALDI-TIMS-SIMSEF-MS workflow.** 1, the sample is imaged in TIMS-MS[1] mode. 2, non-target feature detection is applied in MZmine 3 and 3, the resulting feature list is sorted by ascending area (summed intensity) to search eligible spots for low-intensity precursor ions first. 4, every possible spot where a precursor was detected is evaluated for non-overlapping mobility isolation, spatial criteria for various collision energies, and spectral purity within the precursor m/z isolation window. The resulting precursor lists are saved to one .csv file per spot. 5, the scheduled TIMS-MS[2] spectra are acquired by a Python script and a prototype instrument control software.

list, summarising all possible precursor ions in the dataset. The developed SIMSEF algorithm finds pixels for MS$^2$ experiments (see SIMSEF−Precursor scheduling in MZmine), which are acquired using a custom Python script (see The MS$^2$ acquisition tool and data format). After scheduling, MZmine 3 can directly launch the data acquisition on the acquisition computer. Otherwise, the schedule can be moved to the instrument computer and launched manually. Furthermore, data analysis modules allow evaluation of the acquired fragmentation experiments. Acquiring TIMS-MS$^1$ experiments with a laser spot size smaller than the raster size preserves sample area for subsequent TIMS-MS$^2$ experiments, in which the laser spot size is increased to the raster size, resulting in sufficient analyte ionisation and increased signal-to-noise ratios. The TIMS-MS$^1$ imaging run is set up in flexImaging 7.2, and the imaging run geometry is saved for the subsequent TIMS-MS$^2$ acquisition.

## SIMSEF−precursor scheduling in MZmine

The MZmine 3 import for Bruker *.d/tdf* data was extended to import spot names and coordinates in imaging analyses. A dataset-dependent precursor scheduling algorithm was designed to ensure several quality criteria in the created MS$^2$ spectra, including spectral and spatial considerations. Due to the ion mobility separation and quadrupole switching times in the low millisecond range, multiple precursors are scheduled within a single IMS ramp. The scheduler exploits this to queue multiple precursors in a single imaging pixel. Furthermore, the algorithm considers criteria such as minimum intensity in a pixel, expected isolation purity, distance between MS$^2$ pixels, and ensures enough quadrupole *m/z* switching time. These filters aim to increase the chance for "pure" MS$^2$ spectra, but co-isolation of isobaric and isomeric interferences remains an issue that may be flagged and analysed during downstream processing in MZmine. The SIMSEF algorithm is illustrated in Supplementary Fig. S1. The SIMSEF scheduler works with the results of the MZmine feature detection workflow, which produces mobility-resolved extracted ion images. Every feature is additionally associated with an extracted ion mobilogram and intensities in every pixel[39].

Initially, the MZmine feature table is sorted from the lowest to the highest signal area, i.e., the sum of all signal intensities across the sample. Thereby, low abundant precursors are scheduled first to ensure fragmentation in their most intense pixels. This further increases the chance of acquiring enough MS$^2$ scans for low-abundant compounds only detected in few spots throughout the image. The respective pixels for each precursor are considered from maximum to minimum intensity. Each of these TIMS-MS$^1$ pixels is assessed individually for several quality criteria. Initially, the intensity at which a precursor was detected must exceed an intensity threshold (Minimum MS$^1$ intensity). Then, the scheduler ensures that the mobility window of the precursor does not overlap with already scheduled precursor ions in the same pixel (see Supplementary Fig. S2a). Afterward, spatial limitations are considered to distribute MS$^2$ scans across the sample (see Supplementary Fig. S3). Usually, non-targeted analyses have no way of optimising fragmentation energies for every precursor before the MS$^2$ acquisition, however, this limitation is overcome by scheduling the same precursor in multiple pixels with different, user-defined collision energies. Thereby, the algorithm exploits that the number of pixels in imaging experiments is orders of magnitude higher than the number of detected features. In case multiple MS$^2$ spectra shall be acquired per precursor, the MS$^2$ events of the same collision energy are scheduled with a minimum distance away from each other, which is defined as the Euclidean distance by the user. However, the minimum distance is ignored for MS$^2$ pixels of different collision energies. If a pixel matches the previous criteria, the isolation width of the quadrupole around the precursor *m/z* in that pixel is assessed within the mobility window of the precursor for its expected purity (Minimum purity score) (see Supplementary Fig. S2a). Once a pixel passes all

quality checks, the scheduler adds the precursor to the pixel. This process is repeated until all pixels have been tested or the required number of MS$^2$ spectra have been scheduled for the precursor. The remaining precursors are distributed by the same algorithm, but considering already scheduled MS$^2$ pixels first. After all precursors are scheduled, the necessary information for the acquisition is exported to a project folder. The folder contains the SIMSEF schedule, a precursor list for every MS$^2$ pixel, and collision energy information. IMS-MS imaging datasets with thousands of pixels and millions of mobility-resolved mass spectra are better processed on data processing computers with enough RAM (e.g., 64–128 GB) and multi-core processors. Therefore, the MZmine SIMSEF scheduling can be performed on another computer, and the resulting schedule can be transferred to the data acquisition computer. A graphical preview of scheduled MS$^2$ events for every precursor simplifies the evaluation of the chosen parameters and lets the user decide if satisfactory MS$^2$ coverage is achieved (see Supplementary Fig. S4). The source code is provided in the official MZmine GitHub (https://github.com/mzmine/mzmine3).

## The MS$^2$ acquisition tool and data format

A Python script was developed to run the acquisition list supplied by MZmine 3, utilising a vendor-supplied Python library to control the instrument. The script sets the precursor isolation width, collision energies, and mobility windows and moves the sample stage according to the coordinates and the geometry files stored during the TIMS-MS$^1$ acquisition. The timsControl 4.1 prototype then acquires TIMS-MS$^2$ spectra of the precursor lists, by switching the quadrupole isolation mass along the IMS ramp[34]. The results of one acquired TIMS ramp with 11 precursor ions are compiled in Fig. 2, depicting the extraction of TIMS-resolved precursor-specific MS$^2$ spectra. Rule-based lipid annotation in MZmine annotated phosphatidic acid (PA) 18:0_18:1 based on lipid headgroup and fatty acyl (FA) chain fragments[40,41]. Here, it should be noted that it is not expected that every TIMS-MS$^2$ pixel contains high-quality spectra for every precursor. The applied collision energy may be too high or too low for some molecular species, hence SIMSEF allows scheduling of multiple collision energies, to address this issue.

## Data evaluation in MZmine

Developed MZmine 3 modules now simplify the evaluation of fragment ion spectra from SIMSEF experiments. A MALDI-MS$^2$ grouping module links fragmentation spectra to their corresponding image features. The MS$^2$ grouping relies on the spot names of the imaging analysis, the precursor *m/z*, and the mobility window in which the precursor was isolated. All three criteria must match the MS$^1$ feature. The MS$^2$ scans are extracted for each pixel and then optionally merged into consensus spectra with the same collision energy and across energies, taking the maximum intensity of a signal across all scans (see Fig. 2d). During the spectral merging, even low abundant spectra contribute to the final quality of the TIMS-MS$^2$. Afterwards, existing MS$^2$-based annotation modules in MZmine are available for compound annotation, such as spectral library matching, rule-based lipid annotation, and fragment ion-based formula prediction. Data export and direct interfaces allow integration of the results with open source tools, such as SIRIUS and the GNPS ecosystem for further compound annotation, verification, and molecular networking. Figure 3 demonstrates how MZmine facilitates inspection of raw data and annotation results in its graphical user interface (GUI). Here, the extracted image view is interactively linked to the underlying spectral raw data. Spectral mirror matches can compare two MS$^2$ spectra either between a feature and a spectral library or between two features. In this example, rule-based lipid annotation[41] and spectral matching against the GNPS library, annotated features as glutathione (**a**), phosphatidyl serine (PS) 18:0_18:1 (**b**), and PS 18:0_22:6 (**c**). Rule-based annotation tagged the headgroup neutral loss and both chain fragments for both lipids in

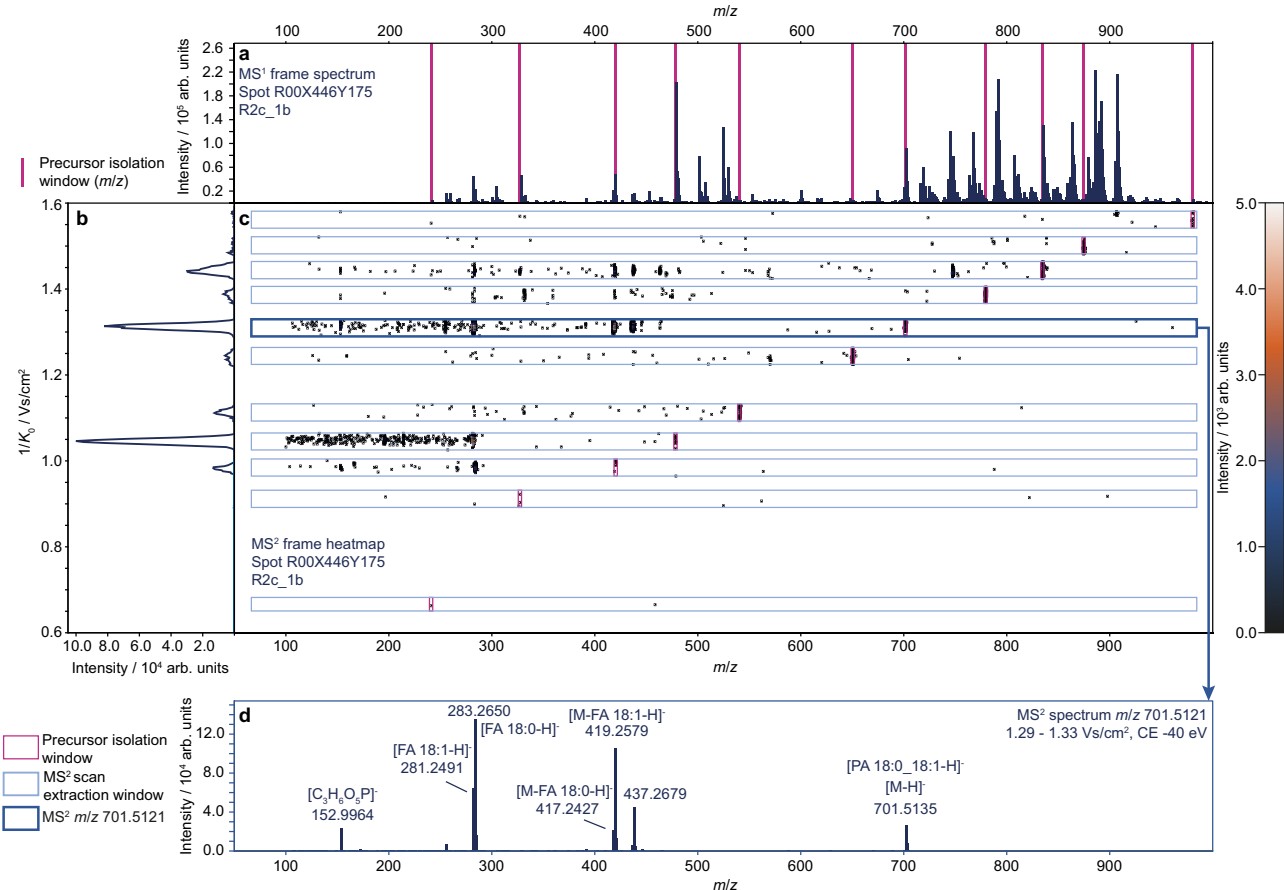

**Fig. 2 | Composition of MALDI-TIMS-MS² frame data with multiple fragmented precursor ions.** SIMSEF scheduled 11 precursor ions (magenta) for **a**, a TIMS-MS1 frame spectrum, here visualised as accumulated spectrum with merged mobility dimension. The corresponding acquired MS² frame visualised as **b**, the total ion mobilogram showing the intensity distribution along the mobility dimension and **c**, the mobility-m/z heatmap of the TIMS-MS² pixel, showing precursor m/z isolation windows (magenta) and ion mobility MS² extraction windows for individual precursors (blue). **d**, an extracted TIMS-MS² scan for the precursor m/z 701.5121 within 1.29 – 1.33 Vs/cm² (bold blue) was annotated with the rule-based lipid annotation module and enriched by manual annotation of fragment signals. The observed headgroup fragment (m/z 152.9964) and signals for chain fragment ions (FA18:0; FA18:1) allow annotation of this precursor as PA 18:0_18:1. The applied collision energy produced high-quality fragment spectra for lipid species between m/z 400 – 850, while being too low for larger and too high for smaller molecules.

accordance to the shorthand notation[42]. Measured CCS values of glutathione (162 Å²), PS 18:0_18:1 (286 Å²) and PS 18:0_22:6 (292 Å²) are within 3% deviation with reported literature values for other IMS analysers (glutathione: 164.24 Å², PS 36:1: 289.4 Å², 279.1 Å²; PS 40:6: 295.4 Å², 284.9 Å²)[43–45] It should be noted that the fatty acid chain composition was not specified in literature, which, including the different IMS setup, could explain the CCS discrepancies. Additional annotations for small metabolites are shown in Supplementary Fig. S5. A composition of all obtained spectral library matches is provided within the Source Data zip file merged_speclibrary_matches.pdf. Additional supporting charts are composed in features_summary.zip contained in the MassIVE dataset (see data availability). GNPS feature-based molecular networks in Fig. 4 and Supplementary Fig. S6 show clustering of compound classes by connecting features by their modified MS² cosine similarity. Figure 4a shows all networks containing 3 or more nodes, with the largest network composed of lipids, while Fig. 4b shows a smaller sub-network of annotated and structurally related metabolites. Furthermore, an imaging specific 'All MS/MS' visualizer shows the extracted ion image and indicates the spots in which an MS² spectrum was acquired. The indicated spots are colour-coded by their collision energy. The MS² spectra are shown on the side of the visualizer, as well as the extracted ion mobilogram of the precursor ion and the mobility isolation window (see Supplementary Fig. S7).

Since SIMSEF scatters fragmentation spectra across the whole tissue instead of using only hotspot regions, the acquired MS² information can be used to identify isomeric or isobaric compound distributions, which were not separated in IMS and m/z dimension (see Supplementary Fig. S8). Additionally, an MZmine module compares the similarity of MS² spectra of the same collision energy for every individual feature. This can help to identify chimeric distributions, if the fragmentation pattern changes within the MS² replicates across the tissue. In the present dataset, 68% of all MS² pairs had an intra-row similarity of ≥0.7 (see Supplementary Fig. S9).

## Performance evaluation

Table 1 shows an overview of the scheduling parameters as Purity score, Quadrupole switch time (ms), and the Minimum distance of MS² pixels. In dataset R2c_a, 1514 features were detected with an intensity of at least 5000 in one spot, making them eligible for scheduling. Method 1 shows the experiment parameters described here (see Online Methods), while methods 2–5 were scheduled for comparison but not queued for data acquisition. In Method 1, a purity score of 0.8 was used to prohibit the scheduling of MS² events that would lead to chimeric MS² spectra. The quadrupole was set to a sufficient switching time of 1.65 ms. The distance between MS² scans was set to 20 pixels, to distribute the MS² spots across the tissue and to avoid over-scheduling in hotspot regions. Methods 2–4 altered

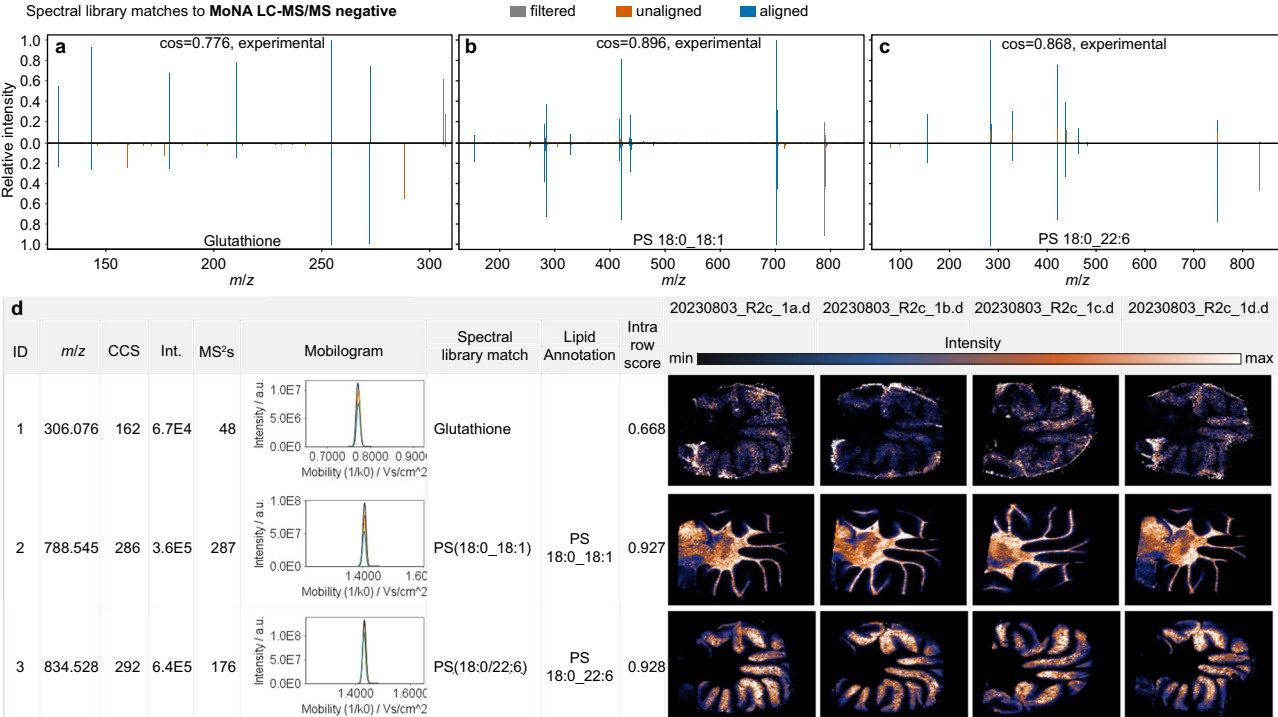

**Fig. 3 | Composition of data and result views in the MZmine GUI. a–c** show spectral library matches to the public MoNA LC-MS/MS negative mode library. Spectral match to a reference spectrum of the metabolite glutathione (**a**), and the phosphatidyl serines PS 18:0_18:1 (**b**) and PS 18:0_22:6 (**c**). Multiple MS² spectra were acquired across the tissue by MALDI-TIMS-SIMSEF. **d** The feature table summarises all features by their properties (e.g., m/z and CCS) and adds interactive charts linked to the underlying data structures. SMART notation for images[48]: Step size: 50 µm, spot size: 30 µm, identification confidence: MSI level 2, resolution: 40,000 FWHM @m/z 1,221, Time 48 min (R2c_1a), 50 min (R2c_1b), 40 min (R2c_1c), 54 min (R2c_1d).

one of these parameters, while method 5 altered all of them, as indicated by the bold cells in Table 1. Method 1 schedules 1396 of the 1514 precursors in 6477 MS² pixels with a total of 28,916 MS² events. A spectra coverage (SC) parameter was calculated from the sum of all scheduled MS² scans per precursor ($N_{scheduled,p}$) and divided by the theoretical maximum number of scheduled MS² scans (Number of precursors ($N_p$) multiplied by the number of spectra ($N_{spectra}$) and collision energies ($N_{CE}$)) (see formula 1).

$$SC = \frac{scheduled\ MS^2\ events}{theoretical\ MS^2\ events} = \frac{\sum_{p=1}^{N_p} N_{scheduled,p}}{(N_p \cdot N_{spectra} \cdot N_{CE})} \quad (1)$$

The spectra coverage describes the ratio of scheduled TIMS-MS² events which pass all quality criteria to the theoretically possible MS² events. For the rat brain dataset R2c_1a, the SC was 69.0% with an average of 4.5 precursors per spot. Reducing the quadrupole switch time from 1.65 ms to 1.00 ms slightly reduces the number of required MS² pixels and increases the number of MS² events, since more precursors are scheduled per spot. However, this can produce chimeric spectra, due to too low switch times which lead to multiple precursors in a single isolation. Reducing the minimum distance of MS² pixels to 5 pixels also increases the number of MS² events, as seen in Method 3. Method 4 decreases the purity score from 0.8 to 0.6, which increases the spectra coverage to 74.4%, while also increasing the number of scheduled precursors, indicating that spectral purity is an influential factor for SIMSEF scheduling. Method 5 combines the previous changes and shows an even higher coverage while requiring fewer spots and increasing the average number of precursors per spot further. Although method 5 would produce more spectra and lead to a higher coverage, the MS² spectra would be of lower purity, distributed closer

together, and contain chimeric isolations. Therefore, method 1 was selected for acquisition.

In sample R2c_1a, 28,916 MS² events were scheduled resulting in 23,444 non-empty scans, i.e., scans with at least one signal above the electronic noise level. Rating the spectral information content, 17,887 (76%) scans contained at least 4 signals and a base peak intensity of at least 1000 (10x noise level). 12,540 (53%) of the TIMS-MS² spectra also had a base peak-to-total ion count ratio of below 0.5, meaning that multiple signals contribute to the overall spectral intensity. This may indicate appropriate quality for metabolite annotation. 83% of features with non-empty MS² scans had at least one MS² spectrum that matched all criteria. A summary for all replicates is shown in Supplementary Table S2. Briefly, the quality of the acquired MS² spectra is reproduced across all datasets. The TIC distribution of all MS² spectra is shown in Supplementary Fig. S10. A combination of rule-based lipid annotation and spectral library matching to the MassBank EU, Lipid BLAST negative ion mode, MoNA, and GNPS reference libraries annotated 208 out of 1514 features in sample R2c_1a (14%). Replicate measurements showed similar annotation rates of 13–15%. Supplementary Fig. S11 shows an exemplary spectrum of a feature annotated as LPE 16:1 by rule-based lipid annotation. The TIMS-MS¹ feature was detected with a maximum intensity of just 9500. Still, the MS² spectra acquired by SIMSEF allowed the annotation of the FA 16:1 fragment ion. This information lacks in the TIMS-MS¹-only imaging data.

## Discussion

We described SIMSEF, a strategy for the dataset-dependent acquisition of MS² spectra in IMS-MS imaging analysis. SIMSEF schedules fragmentation events across an already acquired sample in TIMS-MS¹. This workflow maximises the MS² coverage while exploiting the benefits of TIMS to schedule multiple, mobility separated precursor ions in a single image pixel. Further, the MS² locations are selected by

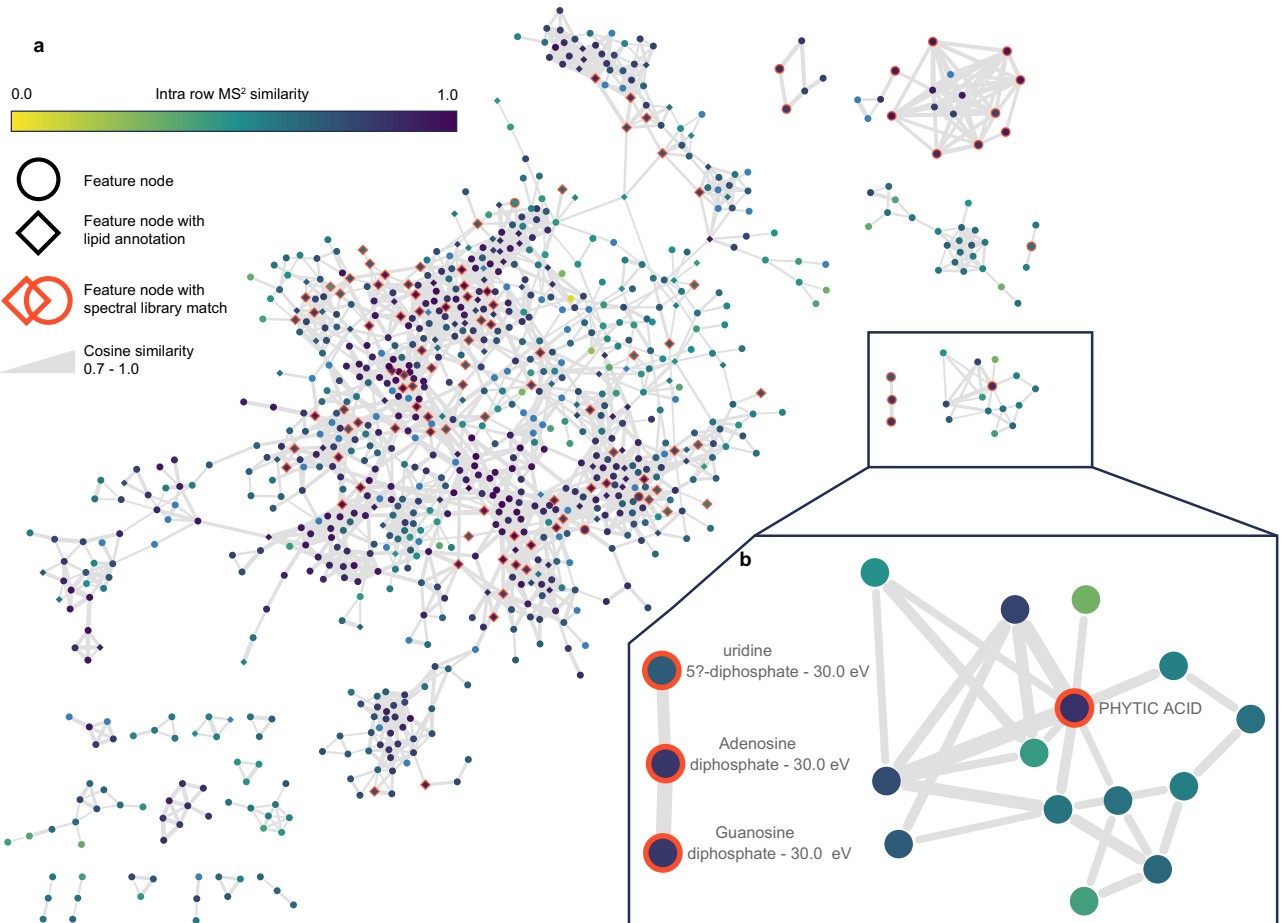

**Fig. 4 | Feature-based molecular networks (FBMN) created from the SIMSEF imaging experiment using the GNPS ecosystem (minimum matched signals 4; minimum cosine similarity 0.7, max cluster size: off).** These networks prove the gained data and annotation quality without the need to acquire additional LC-IMS-MS² data. Nodes describe ion image features with MS² spectra. A single representative, i.e., the most abundant MS² spectrum, was exported for each feature. Edges connect MS² spectra based on their modified cosine similarity (0.7–1.0) which is reflected by the edge weight. Diamond-shaped nodes are annotated by the MZmine rule-based lipid annotation module. Nodes with orange borders are annotated by spectral library matching to the MoNA, MassBank EU, MSDIAL Lipid Blast, or GNPS public libraries. Network **a** shows the lipid sub-network with many connections between annotated lipid species with a single fatty acyl chain substitution. Network **b** shows a small sub-network of annotated metabolites, such as nucleic acid diphosphates and phytic acid. The molecular networking job is available at https://gnps.ucsd.edu/ProteoSAFe/status.jsp?task=9a6f6b34367f4ff69d81c9efe6aedd03.

considering the precursor intensity, spatial distance, and expected isolation purity, to achieve optimal spectral quality and spectral repetition from various image locations. The SIMSEF algorithm was embedded into the open source software MZmine to provide a straightforward and accessible workflow. The acquisition of the scheduled experiments is executed by a Python script, controlling a timsControl 4.1 prototype. Post-acquisition data analysis workflows were implemented in MZmine, assigning the MS² spectra to their TIMS-MS¹ features and allowing a subsequent compound annotation by spectral library matching, rule-based lipid annotation, and the export to other community tools, such as SIRIUS and the GNPS ecosystem. The gained MS² depth enables molecular networking to map the (un) known chemical space and propagate knowledge throughout the spectral networks. Thereby, the SIMSEF workflow in MZmine allows more confident compound annotation for MS imaging analysis, without requiring acquisition of multimodal datasets by LC-IMS-MS and MALDI-IMS-MS. This greatly reduces the experimental complexity of spatial metabolomics studies. The whole SIMSEF workflow is described in the MZmine documentation, including a step-by-step guide (https://mzmine.github.io/mzmine_documentation/workflows/simsef/simsef_workflow.html). SIMSEF improves the MS² coverage and data quality in TIMS-MS imaging, but technical challenges remain in resolving interfering ions that may produce chimeric mass spectra.

## Methods

### Ethical statement

Rat (rattus norvegicus domestica, Wistar strain, female, 58 ± 3 days) samples in this study originated from an animal study (ZH214/16+) approved by the Swiss Federal Veterinary Office Zurich. The study was performed according to the Swiss Animal Welfare Act (TSchG, 2005) and Swiss Animal Welfare Ordinance (TSchV, 2008). Samples were collected for postmortem analysis.

### Materials

Methanol (LC-MS grade) was obtained from VWR International GmbH. N-(1-naphthyl) ethylenediamine dihydrochloride (NEDC) (>99%) was obtained from Carl Roth GmbH + Co. KG. Water was purified by a Milli-Q Academic system (18.2 MΩ cm; 0.2 µm filter; Millipore).

### Sample preparation

Rat brain tissue was cut into sections stored at −80 °C. Parasagittal sections of rat cerebellum were prepared using a CryoStar NX70 (Epredia) in 10 µm thickness and placed on indium tin oxide coated glass slides (70–100 Ω/sq, Sigma Aldrich). Bright field microscopic images were acquired on a BIOREVO BZ-9000 digital microscope (Keyence). A parallel cut of parasagittal cerebellar section was H&E stained (see Supplementary Fig. S12).

**Table 1 | Influence of different SIMSEF parameters on the scheduling of TIMS-MS$^2$ precursors**

| Method | Purity score | Quadrupole switch time / ms | Minimum distance of MS$^2$ pixels/pixel | MS$^2$ pixels | Precursor | MS$^2$ events | Spectra coverage (SC) | Precursors/spot (avg/max) |
|--------|-------|-------|-------|-------|-------|-------|-------|-------|
| 1 | 0.8 | 1.65 | 20 | 6477 | 1396 | 28,916 | 69.0% | 4.5/13 |
| 2 | 0.8 | **1.00** | 20 | 6224 | 1396 | 29,205 | 69.7% | 4.7/16 |
| 3 | 0.8 | 1.65 | **5** | 6508 | 1397 | 29,704 | 70.9% | 4.6/14 |
| 4 | **0.6** | 1.65 | 20 | 6996 | 1503 | 33,528 | 74.4% | 4.8/13 |
| 5 | **0.6** | **1.00** | **5** | 6725 | 1503 | 34,895 | 77.4% | 5.2/17 |

Changed parameters are printed in bold font. The TIMS-MS$^1$ imaging dataset of sample R2c_1a (ramp time 150 ms, MS$^1$ pixels 16,161, scheduling for 6 different collision energies with a replicate number of 5 MS$^2$) was used. A total of 1514 precursor ions had at least one signal above an intensity of 5000 and were thereby eligible for scheduling. TIMS-MS$^2$ scans were only scheduled if the intensity in a pixel exceeded 3000 and 20% of the maximum intensity. The final number of scheduled precursors is lower due to constraints on isolation purity, spatial restrictions, and quadrupole switch time.

NEDC (7 mg/mL) was dissolved in methanol/water (7:3 $v/v$)[46]. The MALDI matrix was applied by a HTX TM-sprayer (HTX Technologies) with the following settings: 10 psi nitrogen pressure, 40 mm nozzle height, 0.120 mL/min flow rate, 75 °C spray temperature, in a 'CC pattern', with 1200 mm/min z-arm velocity, 10 passes, 3 mm track spacing and 0 s drying time.

### SIMSEF workflow
The workflow comprises multiple steps that are described in the next sections. All steps for the MS$^1$ and MS$^2$ data acquisition are outlined in Supplementary Note 1. Parameters for the SIMSEF scheduling algorithm in MZmine are described in Supplementary Note 2. The general step order is:

1. TIMS-MS$^1$ image acquisition
2. Data analysis and SIMSEF scheduling in MZmine
3. SIMSEF-MS$^2$ data acquisition
4. Merging of TIMS-MS$^1$ and MS$^2$ data in MZmine
5. Data analysis and compound annotation

### TIMS-MS$^1$ image acquisition
MALDI-IMS-MS negative ion mode data were acquired on a timsTOF fleX (Bruker Daltonics GmbH & Co. KG) using timsControl 4.1 and flexImaging 7.2. The imaging run was exported from flexImaging 7.2 via "Save run file as…" to make the geometry files available for MS$^2$ measurements.

R2c_1a-d: Imaging data were acquired with a laser field size of 30 μm. The imaging raster size was 50 μm to keep material for subsequent MS$^2$ acquisition (see Supplementary Fig. S13a). Resulting in a final image resolution of 50 μm. The laser frequency was 10,000 Hz with 100 shots and 1 burst per pixel. The laser parameters were set to 'Custom' (Application), a 0% (Power Boost), single (Smart Beam), enabled (Beam Scan), and 26 μm (Scan Range). The mass range was set to 100–1500 $m/z$, the $1/K_0$ range was set to 0.65–1.75 Vs/cm$^2$ and a 150 ms ramp time. Tune parameters were set to 50 V (MALDI plate offset), −70 V (Deflection delta), 500 Vpp (Funnel 1 RF), 0.0 eV (isCID), 350 Vpp (Funnel 2 RF), 350 Vpp (Multipole Vpp), 10 eV (Collision energy), 1100 Vpp (Collision cell RF), 5 eV (Ion energy), 100 $m/z$ (Low mass), 80 μs (Transfer time), 5 μs (Pre pulse storage). TIMS parameters were set to 20 V (Δt1), 120 V (Δt2), −70 V (Δt3), −100 V (Δt4), 0 V (Δt5), −100 V (Δt6), −220 V (Collision cell in). After MS$^1$ acquisition, data analysis was performed in MZmine 3.

### TIMS-MS$^1$ data analysis
An MZmine 3.8 batch configuration is shared through the MassIVE data repository (see Data availability). Briefly, each MS data file (.d folder) was imported to MZmine using the 'Import MS data' module with the advanced import enabled to perform mass detection directly for decreased analysis times. Afterwards, ion images were built for every $m/z$, and subsequently expanded into ion mobility dimension. The extracted ion mobilograms were resolved to features and the resulting feature list was filtered to retain features with a maximum intensity of 5000 or more. The results were used as the input for the scheduling of

MS$^2$ data acquisition using the 'timsTOF SIMSEF imaging scheduler' module.

### SIMSEF scheduling in MZmine
The MS$^2$ schedule was created with a 0.02–0.04 Vs/cm$^2$ precursor mobility window, 1.7 Da precursor isolation width, 5 MS$^2$ spectra for every collision energy, 6 collision energies (20 eV, 30 eV, 40 eV, 50 eV, 60 eV, 70 eV), a minimum distance of 20 pixels between MS$^2$ spectra of the same energy, a minimum absolute intensity of 3000 a.u. or 20%, and a minimum purity score of 0.8 in the respective TIMS-MS$^1$ pixel. An MZmine 3.8 batch configuration is shared through the MassIVE data repository (see Data availability).

### SIMSEF MS$^2$ acquisition
The dataset was acquired using a prototypic version of timsControl 4.1. The scan mode was set to MS/MS and the laser spot size was increased to 50 μm for MS$^2$ acquisitions. Collision RF (800 Vpp) and Transfer time (55 μs) were adapted for ions of lower $m/z$ ions. All other instrument parameters remained unaltered.

### SIMSEF post-analysis in MZmine
An MZmine 3.8 batch configuration is shared through the MassIVE data repository (see Data availability). Briefly, the data were imported and filtered for noise. Image features were grouped with their MS$^2$ scans by the 'Assign MALDI MS$^2$ to features' module. Features were annotated by spectral library matching and rule-based lipid annotation within MZmine. MS$^1$ and MS$^2$ mass tolerances for precursor and fragment ion matching were set to 5 ppm or 0.005 Da and 15 ppm and 0.005 Da, respectively. Afterwards, the processed feature table was exported to execute molecular networking on the GNPS web platform.

### Reporting summary
Further information on research design is available in the Nature Portfolio Reporting Summary linked to this article.

## Data availability
The mass spectrometry imaging raw data generated in this study have been deposited in the MassIVE database under accession code MSV000092935. This dataset also contains the batch configuration and supporting charts in features_summary.zip. Within the Source Data zip file, merged_speclibrary_matches.pdf composes all spectral library match mirror plots. GNPS FBMN molecular networks and annotations are available at https://gnps.ucsd.edu/ProteoSAFe/status.jsp?task=9a6f6b34367f4ff69d81c9efe6aedd03. Source data are provided with this paper.

## Code availability
MZmine modules are available on the official MZmine GitHub (https://github.com/mzmine/mzmine3) under the MIT open source licence. Documentation and a step-by-step guide are available in the official MZmine documentation (https://mzmine.github.io/mzmine_documentation/module_docs/tools_simsef/simsef.html,

https://mzmine.github.io/mzmine_documentation/workflows/simsef/simsef_workflow.html).

The source code for the MS² acquisition tool is available on Zenodo (DOI: 10.5281/zenodo.8009939) and GitHub (https://github.com/SteffenHeu/simsef_py)[47]. Working binaries are available on GitHub (https://github.com/SteffenHeu/simsef_py/releases). Bruker bindings were removed in the source code. To obtain the timsControl 4.1 prototype, researchers may contact their Bruker Daltonics representative.

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

## Acknowledgements

K.K. and U.K. were supported by the Deutsche Forschungsgemeinschaft (DFG) – CRC 1450 – 431460824. R.S. was supported by the Czech Science Foundation (GA CR) grant 21-11563 M. C.B. was supported by the Czech Academy of Sciences PPLZ fellowship number L200552251.

## Author contributions

S.H. conceptualised and implemented the scheduling algorithm, the instrument control script, data evaluation tools in MZmine 3, prepared and acquired MALDI measurements. S.H., A.B. conceptualised the imaging MS2 application. A.B., K.K. prepared thin sections of rat brain tissue. I.D.N., K.K., C.W., C.B. provided test data during development and procedures for sample preparation. A.F. supported S.H. during the Python control script development. R.S. reviewed code, provided conceptual support. A.J., H.R. provided sheep brain tissue during method optimisation. A.J., H.R. provided the rat brain samples. R.S., U.K. supervised S.H. S.H., R.S., wrote the initial manuscript, all authors revised the final manuscript.

## Competing interests

A.B., A.F., I.D.N., A.K. are currently employed at Bruker Daltonics GmbH & Co. KG. The remaining authors declare no competing interests.
