## [Peer Review File · Nature Communications]

Reviewers' Comments:

Reviewer #1:

Remarks to the Author:

In recent years, mass spectrometry imaging (MSI) has emerged as a major platform technology for spatially-resolved analysis of drugs, metabolites, lipids and more – with many application in basic and clinical research as well as in industry. Despite major technological progress in MSI hardware and software, several challenges remain. In this manuscript, Heukeroth et al. address one of the major challenges, namely the lack of automated, systematic on-tissue spatially-resolved MS/MS capabilities (in contrast to few bespoke metabolite fragmentations that are possible in current MSI).

A few years ago, the introduction of trapped ion mobility spectrometry (TIMS)-enabled MS together with powerful new algorithms like Parallel Accumulation–Serial Fragmentation (PASEF) have introduced a very strong new technology into LC-MS-based OMICs (that – needless to say – would be unthinkable without MS/MS capabilities!), which offers an alternative to hybrid ion traps and other technologies for the scientific community. The full potential of this technology in MSI could not yet be fully exploited, since powerful algorithms similar to PASEF and corresponding software implementations (and, hence, systematic MS/MS) have not been available for the scientific community.

To this end, Heukeroth et al. describe a new open source algorithm, “spatial ion mobility-scheduled exhaustive fragmentation (SIMSEF), a dataset-dependent acquisition strategy that augments TIMS-MS imaging datasets with MS2 spectra, where fragmentation experiments are systematically distributed across the sample and scheduled for multiple collision energies per precursor ion”. It is linked to the commonly used open source IT tool mzMINE. The authors exemplify algorithm, workflow and lipid annotation using a brain tissue cryosection.

The authors should be commended for their important contribution to the MSI field.

Nevertheless, the following points should be addressed:

Major Concerns

1. The quality of the brain section that is used as the sole example is very poor. Part of the section is rolled up, another part incl. cortex is ruptured... Moreover, the sectioning plane is unclear. A much better publication-quality example is needed, which can easily be recognized either as a coronal or sagittal brain section (and should be described as such in the Figure legend). Supporting H&E-staining and referencing against the Allen brain atlas is recommended.
2. References should be reconsidered: Reference 3 is a misfit for “increased specificity”, since MALDI-IHC does not require MS/MS. More fitting examples for this can surely be found ranging from hardware to software to statistical solutions. Also, ref. 5 presents impressive speed, but virtually no molecular specificity – which is the essence of this manuscript.
3. Throughout the manuscript it is not always clear if the authors refer to conventional (x, y, m/z) data when referring to data, images or precursors. Or if they refer to TIMS-on data (x, y, CCS, m/z). Please, ensure that this clear in all cases.
4. Acquiring IMS-MS1 data at small laser spot size, but MS2 data at larger spot size to improve sensitivity, is a smart idea. However, how is this reconciled with the notion “expected precursor purity of an isolation window and mobility separation” (line 89)? If the spot size is increased, how is the precursor ion purity evaluated? Based on multiple stitched MS1 pixels?
5. The reader would benefit from having more details on the precursor scheduler in the main manuscript: How are mobility and minimum MS1 intensity assessed by SIMSEF? How are different collision energies chosen? How are minimum distance and minimum purity score calculated? How is data flow handled between the SIMSEF scheduling computer and the conventional data acquisition computer? Do they have to reside in the same lab, or what kind of transfer line is required?

6. The current example in Fig. 2, a PS, is a rather abundant glycerophospholipid. The same is true for the LPE in Fig. 3. It would be important to see 2-3 additional examples covering more challenging lipid classes like gangliosides or non-lipid metabolites to conclude that SIMSEF is a widely applicable tool.

7. More information is needed, in order to grasp the performance of the SIMSEF set-up. For example: For the presented mouse brain section (how many pixels in total?), how long did it take to acquire the IMS-MS1 data? How long to compute the DDA precursor list (steps 2 to 4 in Fig. 1)? How long to acquire the IMS-MS2 data?

8. The same applies to compound annotation: How many database hits for the 4-5 examples (see my 6.) would the user get in database search with only i) m/z data; ii) m/z plus CCS data; iii) m/z plus CCS plus MS/MS data? How many lipids and how many non-lipid metabolites were unequivocally identified using untargeted SIMSEF? This could be done as expansion of lines 220-225.

9. Please, comment on whether SIMSEF works equally well for metabolites that benefit the most from MALDI-2 post-ionization, e.g. HexCer or cholesterol species.

10. Line 256ff.: "A combination of rule-based lipid annotation and spectral library matching to the GNPS reference library annotated 104 out of 1038 features". These were 1038 IMS-MS1 features? Since this is only 10% of the features, please comment on possible causes of what seems to be a rather low annotation rate (e.g. redundant features, false-positive features like matrix peaks etc.). How could annotation rate be improved? Please, discuss. Could other libraries such as HMDB, LipidMaps be used instead of GNPS?

Minor considerations

a. Lines 72-74: "Here, laser capture microdissection is a common approach to sample regions from tissue thin sections for subsequent digestion and LC-MS analysis.". LMD is not yet a common approach in spatial metabolomics. Also, please specify what kind of digestion this refers to, as the reader may infer "trypsin", which is not relevant here.

b. Line 86: Please, ensure clarity, since TIMS-qTOF-MSI experiments are often conducted in TIMS-off mode for increased speed: "First, MALDI-TIMS-qTOF-MS analysis produces MS1 images". If this refers to TIMS-on images, perhaps use "TIMS-MS1 data sets" or "CCS-MS1 data sets" throughout the manuscript to clearly distinguish them from conventional non-TIMS MS1 images.

c. Line 92: Please introduce GNPS and SIRIUS abbreviations.

d. Line 107: should this read "restrains sample area"?

e. Line 138: perhaps "the same precursor"? How are the "different collision energies" chosen?

f. Line 157: Figure S4Error! Reference source not found.).

g. Since a vendor is very much involved here: Is every component described in the manuscript truly open source, or what restrictions may apply? For example, what is the availability status of the timsControl 4.1 prototype (line 163)?

Reviewer #2:

Remarks to the Author:

This manuscript by Heuckeroth et al describes the creation of a new workflow for mass spectrometry imaging data sets. The team has created "spatial ion mobility-scheduled exhaustive fragmentation (SIMSEF)" this is a dataset-dependent acquisition strategy the leverages the MS1 and trapped ion mobility spectra to generate a program for targeted MS2 acquisition at multiple

collision energies across a tissue section. The team rightly argues that MALDI-based mass spectrometry imaging has historically relied on high resolution MS1 data to annotate the datasets, with many research groups often times employing various chemical extractions or laser capture microdissection to extract analytes for orthogonal analyses. Overall, within the MSI community, this tool will be immensely helpful, however, some information and comparisons regarding the resulting data may be expanded to better highlight the utility. Additionally, all the data appears to all be collected from a single tissue of a rat brain with one replicate, which may somewhat misrepresent the reproducibility of the methodology. Below please find critiques which may be helpful to the authors. Overall, this workflow has immense potential.

Major

1. Results section, line 104 Can the authors estimate how long the time would be for processing the MS1 dataset to acquisition of the MS2 dataset? Often for LC-MS/MS this processing can be lengthy, and additionally the combination of MS1 and IMS across the MSI dataset tends to yield vary large datasets, so some considerations for sample and matrix stability within the source may need to be addressed since it is unclear how long it would take to process the data prior to the MS2 acquisition.
2. Regarding Figure 2, it is unclear where this is meant to direct the reader with the choices in the figure legends, the pink and blue boxes are easy to identify but not the grey. For some of the lower IM values with the precursor isolation window, it almost appears that no signal is present. Would it be possible to add what the pixel looked like for the ion image this would have been generated from to better contextualize the resulting figure?
Additionally the SMART data reporting would be excellent to include along with any images displayed in the main text and the SI figures for transparency of this method:
<https://pubmed.ncbi.nlm.nih.gov/36740651/>
3. Page 6, line 195, the presented range of CCS values seems vague given that the position of the double bond is not known, is there a chance this is multiple isomers? That may be important to clearly delineate a continued short coming to identification of some compound classes.
4. Regarding Figure S5, Can the authors estimate how many MS2 spectra yield good quality based on the data presented, the number of arrows seems low? Alternatively, how many features are in the MS1 dataset? It would be nice to make comparisons to MS 1 features, and then features that yield meaningful MS/MS spectra, followed by levels of annotation. This might provide the more general reader a greater grasp of the complexity of the MSI datasets.
5. It is slightly concerning that all this analysis appears across an N=1 experiment. It might be helpful in this evaluation to include consistency across multiple samples to better gauge effectiveness across different experiments.

Minor

1. Page 2, lines 40-42 This feels like the authors specifically mean to refer to MALDI or LDI based imaging approaches, for DESI/nanoDESI, LAESI, IRMALDESI this is simply not the case. MS/MS can be, and is collected as a part of these workflows.
2. Results section, line 108-109 This is more of a comment, depending on the desired spatial resolution, a simple offset of the laser is likely to result in the same results without any over or under sampling.
3. Page 4, line 128 "lowest to highest area" This might be better qualified as peak intensity given that this could be mistaken for MSI area*
4. Is Figure S8 the summary of all the features or just the lipid based subset? This seems very low compared to what could be achieved via LC-MS/MS of the extracted tissue. Again this speaks to how many MS/MS pixels from an imaging run may yield actionable information.
5. Page 10, line 301 I assume this is negative mode based on the matrix selection, but it should be expressly stated.

Reviewer #3:

Remarks to the Author:

This manuscript, entitled "On-tissue dataset-dependent MALDI-TIMS-MS2 bioimaging", aims to develop a type of DDA for imaging mass spectrometry. The objective is to collect MS/MS in a more strategic fashion, thereby enhancing the number of ions that can be identified in an experiment.

The authors first collect MS1 data, essentially at every other spot of the tissue. They then process the data using a custom script they wrote (integrated with MZmine 3). The software selects which ions should be fragmented where. They then acquire a second round of imaging data using MS2 on the spots of tissue remaining.

I have two major questions related to the underlying premise put forward by the authors.

1. The authors argue that the number of compounds identified in timstof imaging is limited by a paucity of DDA modes. At least in our experiences, that has not really been the case. For us, the biggest limitation to identifying compounds is MS/MS sensitivity (especially when the tims is on) and an inability to mass resolve different molecular species. Simply put, no matter how the data is collected, the MS/MS is not high quality enough to substantiate high confidence identifications in most cases. I do not know that DDA helps that issue.

Just because one collects MS/MS on an ion, doesn't mean that it is above noise enough to merit a confident identification. If the MS/MS data cannot lead to a confident identification, then is it worth collecting? How many of the ~1100 ions for which MS/MS was collected led to confident identifications (the authors say they "annotated" 104 features but that term is often used to indicate low confidence assignments, which we can already get from MS1 data alone)? The MS/MS data should be shown and the identities should be included in some sort of table, together with MSI confidence scores. Given that the main argument is that DDA imaging leads to more identifications, it would be helpful to show tables of identifications with and without the DDA approach (together with confidence scores).

2. The underlying premise of DDA is that you only have to collect MS/MS data on an ion one time. After it is identified from that MS/MS spectrum, the ion can then be annotated in other mass spectra. In LC/MS, the assumption is that an ion with a given m/z value across a chromatographic peak represents the same molecule. Or, in DDA, that an ion with the same accurate mass and retention time in different samples is the same molecule. While not always formally true, it is generally a good assumption (unless an ion elutes in the void volume). This is because the combination of LC retention times and accurate mass is specific. In imaging, however, the RT domain is lost. Now, one has to make conclusions about whether an ion at different locations of a tissue is the same or not based only on accurate mass. That assumption is precarious. Figure 9 c-d is a good example. The same ion is fragmented at different locations of the tissue, but the MS/MS patterns are clearly different.

So how does one know when the accurate mass is likely to be the same compound or a different compound? The authors suggest using pixel distances, but I have reservations about this logic. Imagine that an ion with an arbitrary accurate mass is detected at two different locations. In theory, MS/MS could be collected at one spot for identification. One could then hypothesize that ions at the second spot have the same identity. But how do we know that is true if we do not collect MS/MS? I'm not sure distance is a good proxy for assessing the probability of such an event. In heterogeneous tissues, a few micron shift could be sufficient to survey a different cell type. I actually think it's more likely that ions from distant cells of the same type are more likely to have the same identity than ions from neighboring cells of different cell types.

One way to potentially address this would be to collect MS/MS data for all pixels of an image. Then demonstrate that it looks the same within 20 pixels (or whatever distance threshold was used) but that it changes at further distances. It would be necessary to demonstrate this for multiple classes of compounds in different types of tissues. Without further support, the assumption that an m/z value within 20 pixels is the same molecule but that the same m/z value outside 20 pixels is not appears to be totally arbitrary.

A few additional points:

3. The method for assessing chimeric MS/MS data is not complete. In many cases, two molecules will be similar enough in m/z that they cannot be mass resolved by the instrument even in MS1 mode (particularly when the resolution of the timstof is only ~40k). In this case, they will appear to be a single peak in MS1 data. MS/MS is needed to assess whether the composition of ions

contributing to the peak is indeed the result of a single precursor. It's also possible that the peak is pure in the MS1 scan but then chimeric in the adjacent MS2 data, which is a few microns away. These possibilities are not taken into account.

4. The MS1 data is processed off-line using a python script. For adjacent pixels, how much time lapses between the MS1 and MS2 experiments? I am wondering whether NEDC might undergo sublimation during that time, which would complicate the results considerably as it would change the molecular profile of the tissue.

5. The authors mention the use of a "prototype instrument control software". Please clarify if this can be readily implemented by other people or what would be needed to use it on a Bruker tims systems as well as non-Bruker instruments.

6. Why did the authors use NEDC as a matrix?

Minor comments:

Figure 2: Please clarify what "manually enriched" means and potentially add that the serine headgroup is observed as neutral loss.

Line 182: Does "consensus spectra" mean averaged spectra?

Line 238: should be "in Method 3. Method 4 decreases..."

Line 306: Is the raster size for MS1 really 50 μm ? Given the description it seems very large, especially since later for MS2 the laser spot size was increased to 50 μm .

I recommend calling out supplemental figures in order (Figure S7 is currently not referenced in the main text).

Figure S1: Make sure that all options are logical. There are a few points in the flow diagram where the other option (Yes or No) is not accounted for. Some also seem to have an answer different from yes or no (eg, "Determine which collision energy has the fewest MS2 pixel for feature").

Figure S1: typo in "Find exsisting MS2 pixels"

Figure S4: "Minimum chimerity score" seems confusing. Should it be "Maximum chimerity score"?

Point-to-Point Response

Reviewer #1 (Remarks to the Author):

In recent years, mass spectrometry imaging (MSI) has emerged as a major platform technology for spatially-resolved analysis of drugs, metabolites, lipids and more – with many application in basic and clinical research as well as in industry. Despite major technological progress in MSI hardware and software, several challenges remain. In this manuscript, Heukeroth et al. address one of the major challenges, namely the lack of automated, systematic on-tissue spatially-resolved MS/MS capabilities (in contrast to few bespoke metabolite fragmentations that are possible in current MSI).

A few years ago, the introduction of trapped ion mobility spectrometry (TIMS)-enabled MS together with powerful new algorithms like Parallel Accumulation–Serial Fragmentation (PASEF) have introduced a very strong new technology into LC-MS-based OMICs (that – needless to say – would be unthinkable without MS/MS capabilities!), which offers an alternative to hybrid ion traps and other technologies for the scientific community. The full potential of this technology in MSI could not yet be fully exploited, since powerful algorithms similar to PASEF and corresponding software implementations (and, hence, systematic MS/MS) have not been available for the scientific community.

This is an accurate overview of the developments in the mass spectrometry market. The TIMS-TOF-MS indeed offered an alternative to the dominant FTMS instruments. The fact that TIMS accumulates ions makes it ideal for fragmentation analysis and PASEF has been one of the key drivers here. We initially expected that something similar would be available for MALDI-TIMS-MS2 data acquisition soon. However, our SIMSEF algorithms are the first of their kind and leverage the special characteristics of imaging analyses with spatial coordinates. You have precisely captured the needs and limitations that we aimed to address.

To this end, Heukeroth et al. describe a new open source algorithm, “spatial ion mobility-scheduled exhaustive fragmentation (SIMSEF), a dataset-dependent acquisition strategy that augments TIMS-MS imaging datasets with MS2 spectra, where fragmentation experiments are systematically distributed across the sample and scheduled for multiple collision energies per precursor ion”. It is linked to the commonly used open source IT tool mzMINE. The authors exemplify algorithm, workflow and lipid annotation using a brain tissue cryosection.

The authors should be commended for their important contribution to the MSI field.

We appreciate the reviewer’s valuable feedback and insightful suggestions. The summary and conclusion accurately capture our motivation and goals. The points raised prompted us to enhance the workflow in MZmine with new developments. Moreover, we extended our analysis to four samples to demonstrate robustness based on summary statistics for the data acquisition and analysis. Regarding annotations, we implemented an automatic reporting module in MZmine and provided comprehensive plots in the supplementary file: *features_summary.zip*. All the following points are addressed in the point-to-point response.

Nevertheless, the following points should be addressed:

Major Concerns

1. The quality of the brain section that is used as the sole example is very poor. Part of the section is rolled up, another part incl. cortex is ruptured... Moreover, the sectioning plane is unclear.

A much better publication-quality example is needed, which can easily be recognized either as a coronal or sagittal brain section (and should be described as such in the Figure legend). Supporting H&E-staining and referencing against the Allen brain atlas is recommended.

We revised the manuscript with a new dataset. SIMSEF was now executed on 4 parallel, parasagittal sections of rat brain cerebellum. In addition, H&E staining was executed on another parallel section and included in the supplementary material.

Figure S13: H&E staining of a parasagittal section of rat brain cerebellum. The section was parallel to the sections used for MALDI-TIMS-MS Imaging.

2. References should be reconsidered: Reference 3 is a misfit for “increased specificity”, since MALDI-IHC does not require MS/MS. More fitting examples for this can surely be found ranging from hardware to software to statistical solutions. Also, ref. 5 presents impressive speed, but virtually no molecular specificity – which is the essence of this manuscript.

We agree with the reviewer. As reference 3 was more specific to proteomics we changed to the following paper that combines MALDI2-TIMS-TOF-MS for the separation of lipids (<https://pubs.acs.org/doi/10.1021/acs.analchem.0c01747>). This outlines how TIMS can separate lipid species, enhancing the specificity of the method. Also MALDI2 boosts different ions leading to the detection of specific ion species for compound classes.

Regarding reference 5, we share the concern that the increased speed comes at a cost. We revised the text and point to these limitations. Also, we added another reference for increased acquisition speeds: <https://pubs.acs.org/doi/10.1007/s13361-018-2078-8>

The introduction now reads:

In recent years, advances in instrumentation led to higher spatial resolution,² increased specificity,³ boosted sensitivity,⁴ and increased sample throughput.^{5,6} These characteristics are linked and influence each other, for example, higher spatial resolutions require efficient ionisation and increased sensitivity. A major bottleneck in many MS imaging studies remains the compound annotation, which often relies on accurate mass only, due to missing fragmentation data (MS²).^{2,6,7}

3. Throughout the manuscript it is not always clear if the authors refer to conventional (x, y, m/z) data when referring to data, images or precursors. Or if they refer to TIMS-on data (x, y, CCS, m/z). Please, ensure that this clear in all cases.

Thank you for pointing this out. We changed the relevant sections to TIMS-MS1 instead of MS1. All datasets were acquired with TIMS separation on for this paper.

4. Acquiring IMS-MS1 data at small laser spot size, but MS2 data at larger spot size to improve sensitivity, is a smart idea. However, how is this reconciled with the notion “expected precursor purity of an isolation window and mobility separation” (line 89)? If the spot size is increased, how is the precursor ion purity evaluated? Based on multiple stitched MS1 pixels?

Each MS1 raster position of the 50 µm x 50 µm raster is associated with a specific coordinate. All MS1 pixels are assessed by the SIMSEF scheduling algorithm individually. The MS2 pixels are acquired on the same coordinate map, but with a bigger laser spot size. The expected purity of the MS2 is extrapolated from the MS1 pixel only.

We added this sentence to the results section to clarify this process:

The respective pixels for each precursor are considered from maximum to minimum intensity. Each of these TIMS-MS1 pixels is assessed individually for several quality criteria.

and modified this part in the results section:

If a pixel matches the previous criteria, the isolation width of the quadrupole around the precursor m/z in that pixel is assessed within the mobility window of the precursor for its expected purity (Minimum purity score) (see Figure S2a)

It would be possible to include a multi-pixel approach by assessing all surrounding pixels. However, this would increase the processing time, because the purity assessment is one of the most time-consuming tasks of the scheduling. Therefore, it is executed as the last step after a pixel passes all other criteria.

5. The reader would benefit from having more details on the precursor scheduler in the main manuscript: How are mobility and minimum MS1 intensity assessed by SIMSEF? How are different collision energies chosen? How are minimum distance and minimum purity score calculated?

The mobility window is derived from the MS1 feature detection algorithm in MZmine. MZmine offers many more filtering options to reduce the number of features. Intensities of each pixel are considered individually and a mobilogram is built for every precursor/feature. In the MZmine feature table, every TIMS-MS image feature contains information on the intensity in every pixel and therefore the respective scan. Thus, the minimum intensity is checked in every scan.

The collision energies are defined by the user as a list of energies and a number of replicates per energy. The minimum distance is the Euclidean distance between two points (unit = pixel).

The expected purity is derived by summing the intensity of the precursor ion and its ¹³C isotopic signals in the respective mobility and mass isolation window in the TIMS-MS1 pixel and dividing it by the intensity of all signals in that same window. Precursor + isotopes / total.

We adapted the results section with the following additions:

The SIMSEF scheduler works with the results of the MZmine feature detection workflow. The workflow produces mobility-resolved extracted ion images. Every feature is additionally associated with an extracted ion mobilogram and intensities in every pixel.[cite biotech paper]

[..]

The respective pixels for each precursor are considered from maximum to minimum intensity. Each of these TIMS-MS1 pixels is assessed individually for several quality criteria.

[..]

In case multiple MS² spectra shall be acquired per precursor, the MS² events of the same collision energy are scheduled with a minimum distance away from each other, which is defined as the Euclidean distance by the user. However, the minimum distance is ignored for MS² pixels of different collision energies. If a pixel matches the previous criteria, the isolation width of the quadrupole around the precursor m/z in that pixel is assessed within the mobility window of the precursor for its expected purity (Minimum purity score) (see Figure S2a).

Additionally, we adapted the caption of Figure S2 to further highlight that the purity is assessed only in the mobility window of that precursor.

Figure S 2: Spectral scheduling procedure of the SIMSEF algorithm. The SIMSEF scheduler assesses multiple spectral criteria, before scheduling a precursor for MS² acquisition in a particular spot. a, the mobility window of the precursor is examined regarding overlaps with previously scheduled precursors. In case an overlap is detected, the precursor will not be scheduled at this spot. b, the spot selection for precursor ions assesses the expected purity of the quadrupole m/z isolation window within the mobility range of the precursor. The pixel will only be considered if the intensity of the precursor exceeds the intensity threshold (magenta) and passes a spectral purity test in the mobility-resolved MS¹ spectra it was detected in.

How is data flow handled between the SIMSEF scheduling computer and the conventional data acquisition computer? Do they have to reside in the same lab, or what kind of transfer line is required?

The data can be analysed on the data acquisition computer if it is powerful enough. Then the acquisition can be started directly within the scheduling. Otherwise, the acquisition scheduling files can be transferred from any computer via network or USB drives. We clarified this in the main manuscript.

Therefore, the MZmine SIMSEF scheduling can be performed on another computer, and the resulting schedule can be transferred to the data acquisition computer.

6. The current example in Fig. 2, a PS, is a rather abundant glycerophospholipid. The same is true for the LPE in Fig. 3. It would be important to see 2-3 additional examples covering more challenging lipid classes like gangliosides or non-lipid metabolites to conclude that SIMSEF is a widely applicable tool.

In the new datasets with four examples, we adjusted the acquisition method to also cover a lower m/z range to demonstrate the annotation of several non-lipid metabolites. We added glutathione to the main manuscript and provided spectral matches of other metabolites (Adenosine diphosphate and Phytic acid) in the supplementary information. Additionally, we composed a supplementary file that contains additional

plots and library matches, the top-two library matches of every annotated feature. All this is composed in the *features_summary.zip* supplementary file. This option was added as a new module in response to this but also comments from the other reviewers, also asking for more validation.

Further, we'd like to note that SIMSEF only depends on the underlying ionisation technique and sufficient intensity for precursor ions to be selected.

We adapted Figure 3 to the new dataset and captured three example annotations: 3 spectral library matches. The two lipid species were additionally verified by rule-based lipid annotation:

Figure 3: Composition of data and result views in the MZmine GUI. **a**, **b**, and **c** show spectral library matches to the public MoNA LC-MS/MS negative mode library. Spectral match to a reference spectrum of the metabolite glutathione (**a**), and the phosphatidyl serines PS 18:0_18:1 (**b**) and PS 18:0_22:6 (**c**). Multiple MS2 spectra were acquired across the tissue by MALDI-TIMS-SIMSEF. **d**, the feature table summarises all features by their properties (e.g., m/z and CCS) and adds interactive charts linked to the underlying data structures. SMART notation for images: Step size: 50 µm, spot size: 30 µm, identification confidence: MSI level 2, resolution: 40 000 FWHM @m/z 1221, Time 48 min (R2c_1a), 50 min (R2c_1b), 40 min (R2c_1c), 54 min (R2c_1d)

Additional annotations for small metabolites are shown in Figure S7. A composition of all obtained spectral library matches is provided in the supplementary files (*features_summary.zip*)

7. More information is needed, in order to grasp the performance of the SIMSEF set-up. For example: For the presented mouse brain section (how many pixels in total?), how long did it take to acquire the IMS-MS1 data? How long to compute the DDA precursor list (steps 2 to 4 in Fig. 1)? How long to acquire the IMS-MS2 data?

This part was indeed short and we added a table summarising the performance of SIMSEF and MZmine to the supplementary material.

Briefly, the MS1 data acquisition took 39-53 minutes. The processing time for MS1 data was between 9 and 14 min, while the SIMSEF scheduling required roughly 1-2 min. The MS2 spectra were acquired at a rate of roughly 2000 MS2 pixels per hour with multiple MS2 spectra per pixel. Resulting in a total acquisition time of about 3 - 3.5 h for 6000 - 7000 MS2 spectra. Here, 6 collision energies and 5 spectra per collision energy were scheduled per precursor. The MS2 acquisition time can be shortened substantially, by more rigorous pre-filtering of the feature table, using less collision energies or fewer MS2 spectra per energy.

Table S 1 Overview of time and feature statistics of four SIMSEF measurements on parallel tissue thin sections.

	R2c_1a	R2c_1b	R2c_1c	R2c_1d
MS1 pixels	16,161	16,765	13,329	17,984
Acquisition time / min	48.3	50.1	39.9	53.7
Feature detection time / min	9.3	12.0	9.6	14.3
MS1 features	1514	1652	1722	1769
SIMSEF scheduling time / min	1.0	1.8	1.2	2.3
MS2 acquisition time / min	186.0	206.0	205.0	221.0
MS2 pixels	6477	6835	6722	7192
Acquired MS2 pixels per hour / 1/h	2089.4	1990.8	1967.4	1952.6
Individual MS2 spectra per hour / 1/h	9327.7	8667.7	9180.3	8733.9

8. The same applies to compound annotation: How many database hits for the 4-5 examples (see my 6.) would the user get in database search with only i) m/z data; ii) m/z plus CCS data; iii) m/z plus CCS plus MS/MS data? How many lipids and how many non-lipid metabolites were unequivocally identified using untargeted SIMSEF? This could be done as expansion of lines 220-225.

Annotation and confidence reporting is really an outstanding challenge for our community. We think one of the main agreements is that MS2 data is the gold standard for annotation in mass spectrometry-based studies. With SIMSEF we aim to enable smart scheduling of MS2 fragmentation events for TIMS-MS imaging users and aim to further improve the quality of acquired MS2 spectra. To validate the performance, we defined various spectral quality metrics. We added Supplementary Table S2 as a summary of the number of “quality” MS2 spectra and TIMS-MS imaging features extracted from all four datasets:

Table S 2: Summary of calculated quality criteria across four replicate SIMSEF experiments. BPI describes the base peak intensity.

	R2c_1a	R2c_1b	R2c_1c	R2c_1d
Features – total	1,514	1,652	1,722	1,769
Features – scheduled (matching SIMSEF criteria)	1,396	1,514	1,560	1,640
MS² – total	28,916	29,759	31,366	32,170
MS² – non-empty scans	23,444	24,813	26,076	25,705
MS² – at least 4 signals and BPI 10x S/N	17,887	19,126	19,537	19,599
“good” MS² – all the above and BPI / TIC < 0.5	12,540	13,481	13,555	13,867
Features – at least one non-empty MS²	1,329	1,426	1,556	1,460
Features – at least one “good” MS²	1,105	1,219	1,262	1,236
Features – “good” MS² / Features with non-empty MS²	83%	85%	81%	85%
Spectral library matches	100	121	118	101
Lipid annotation	192	223	208	212
Annotated features	208	255	238	232
Annotation rate	14%	15%	14%	13%

All top 2 spectral library matches per feature were exported to pdf files together with their spectral mirror matches as images. The metabolomics standards initiative MSI considers spectral library matches without reference standards a Level 2 annotation. (Still there are many possible isomers that may not be captured in the open public spectral libraries that may exhibit similar fragmentation patterns).

There are great projects that aim to collect experimental CCS values and predict CCS values (e.g., AllCCS, CCSBase, PubChem lite). However, their coverage is still low and would constrain the search space too much for non-target analysis. They may help in verifying and pursuing full elucidation of statistically significant features, though. As CCS lowers the number of potential isobaric interferences, it can be used as an additional annotation dimension. Optimally, a reference standard is measured for comparison on the same system. MZmine can help by extracting this reference data to spectral libraries with CCS values.

9. Please, comment on whether SIMSEF works equally well for metabolites that benefit the most from **MALDI-2 post-ionization**, e.g. HexCer or cholesterol species.

Generally, SIMSEF is ionisation-agnostic. In another project, SIMSEF was also tested with direct laser desorption ionisation (LDI) timsTOF-MS and could, in general, be used with DESI and other ionisation techniques if compatible with the timsTOF-MS platform. Ionisation efficiency and ion yield are most important for the MS2 acquisition.

Our instrument does not have a MALDI-2 laser, however, we tested the workflow on a MALDI-2 instrument with the Dreisewerd/Soltwisch group with similar success.

10. Line 256ff.: “A combination of rule-based lipid annotation and spectral library matching to the GNPS reference library annotated 104 out of 1038 features”. These were 1038 IMS-MS1 features? Since this is only 10% of the features, please comment on possible causes of what seems to be a rather low annotation rate (e.g. redundant features, false-positive features like matrix peaks etc.). How could annotation rate be improved? Please, discuss. Could other libraries such as HMDB, LipidMaps be used instead of GNPS?

Annotation rates in untargeted MS are usually low between 1-10 %, (<https://doi.org/10.1007/s11306-022-01963-y>) depending on the feature finding parameters and the tested samples, they may be higher. The reported 13%-15% annotation rate is based on MS2 annotations only. For spectral library matching, this means MS2 spectra need to contain and match to at least 4 m/z signals (excluding the precursor m/z) and have a cosine similarity greater than 0.7. The low availability of reference spectra is a limiting factor for high-quality annotations. Therefore, we pooled spectral libraries from the largest open spectral libraries MassBank of North America (MoNA), MassBankEU, LipidBlast, and GNPS. Here, GNPS is one of the largest open MS databases and contains around 20,000 unique structures and 600,000 MS2 spectra. MZmine supports annotation based on MS1 only (m/z and/or CCS) but its use is discouraged as the annotation ranges are too wide, leading to many wrong annotations. More sophisticated approaches apply rule-based lipid annotation or SIRIUS compound annotation for small molecules. MZmine supports the whole SIRIUS compound and compound class annotation pipeline through open data exchange formats. SIRIUS uses compound databases like HMDB and PubChem coupled with machine learning models and provides filtering options for bio and general-purpose compound databases. In our view, spectral library matching, rule-based lipid annotation, and in-silico tools are currently great complementary strategies for compound annotation.

We added Supplementary Table S2 to summarise the annotated features and annotation rate.

Minor considerations

a. Lines 72-74: “Here, laser capture microdissection is a common approach to sample regions from tissue thin sections for subsequent digestion and LC-MS analysis.”. LMD is not yet a common approach in spatial metabolomics. Also, please specify what kind of digestion this refers to, as the reader may infer “trypsin”, which is not relevant here.

We acknowledge that LMD is not widely applied in mass spectrometry imaging. Therefore, we modified the statement to emphasise its potential for sampling of regions. Moreover, we agree that digestion is a very specific term and we replaced it with extraction which is more general. The text in the introduction now reads:

Comprehensive spatial metabolomics studies acquire complementary datasets by LC-MS² and MS¹ imaging, without fragmentation analysis. Here, laser capture microdissection enables sampling of regions from tissue thin sections for subsequent extraction and LC-MS analysis.

b. Line 86: Please, ensure clarity, since TIMS-qTOF-MSI experiments are often conducted in TIMS-off mode for increased speed: “First, MALDI-TIMS-qTOF-MS analysis produces MS1 images”. If this refers to TIMS-on images, perhaps use “TIMS-MS1 data sets” or “CCS-MS1 data sets” throughout the manuscript to clearly distinguish them from conventional non-TIMS MS1 images.

We agree this was not clear enough, and we rephrased the text. All measurements were conducted with TIMS separation. The text now reads TIMS-MS1 images to signal both separation techniques.

c. Line 92: Please introduce GNPS and SIRIUS abbreviations.

We are now introducing GNPS as Global natural product social molecular networking but SIRIUS is just the name of a tool or more precisely now a whole tool suite that encompasses SIRIUS, CSI:FingerID, CANOPUS, etc.

d. Line 107: should this read “restrains sample area”?

This sentence was ambiguous. We changed it to the following in the main text:

Acquiring TIMS-MS¹ experiments with a laser spot size smaller than the raster size preserves sample area for subsequent TIMS-MS² experiments, in which the laser spot size is increased to the raster size, resulting in sufficient analyte ionisation and increased signal-to-noise ratios.

e. Line 138: perhaps “the same precursor”? How are the “different collision energies” chosen?

Thanks for the thorough read. We changed the text to better address this concept of replication of MS² with different user-defined collision energies. The main text now reads:

Usually, non-targeted analyses have no way of optimising fragmentation energies for every precursor before the MS² acquisition, however, this limitation is overcome by scheduling the same precursor in multiple pixels with different, user-defined collision energies.

f. Line 157: Figure S4Error! Reference source not found.).

Good catch. Nothing was missing here and this must have happened in the final file conversions.

g. Since a vendor is very much involved here: Is every component described in the manuscript truly open source, or what restrictions may apply? For example, what is the availability status of the timsControl 4.1 prototype (line 163)?

The SIMSEF scheduling algorithm as shown in the preview, is completely open-source and implemented in MZmine. The SIMSEF task with all logic can be found here: https://github.com/mzmine/mzmine3/blob/master/src/main/java/io/github/mzmine/modules/tools/timstofmal_diacq/imaging/SimsefImagingSchedulerTask.java

The data acquisition Python script is open-source and available on GitHub https://github.com/SteffenHeu/simsef_py. A working binary with all dependencies is available for download there. Only the vendor library (Python wheel) had to be removed from the source code.

The timsControl 4.1 prototype is provided by Bruker Daltonics through the Bruker contact of the respective instrument owner. Previously, this was only described in the MZmine documentation page of the workflow. Now, we added this to the code availability section.

Code availability

MZmine modules are available on the official MZmine GitHub (<https://github.com/mzmine/mzmine3>) under the MIT open source licence. Documentation and a step-by-step guide are available in the official MZmine documentation (https://mzmine.github.io/mzmine_documentation/module_docs/tools_simsef/simsef.html, https://mzmine.github.io/mzmine_documentation/workflows/simsef/simsef_workflow.html).

The source code for the MS² acquisition tool is available on Zenodo ([DOI: 10.5281/zenodo.8009939](https://doi.org/10.5281/zenodo.8009939)) and GitHub (https://github.com/SteffenHeu/simsef_py). Working binaries are available on GitHub (https://github.com/SteffenHeu/simsef_py/releases). Bruker bindings were removed in the source code. To obtain the timsControl 4.1 prototype, researchers may contact their Bruker Daltonics representative.

Reviewer #2 (Remarks to the Author):

This manuscript by Heuckeroth et al describes the creation of a new workflow for mass spectrometry imaging data sets. The team has created “spatial ion mobility-scheduled exhaustive fragmentation (SIMSEF)” this is a dataset-dependent acquisition strategy the leverages the MS1 and trapped ion mobility spectra to generate a program for targeted MS2 acquisition at multiple collision energies across a tissue section. The team rightly argues that MALDI-based mass spectrometry imaging has historically relied on high resolution MS1 data to annotate the datasets, with many research groups often times employing various chemical extractions or laser capture microdissection to extract analytes for orthogonal analyses. Overall, within the MSI community, this tool will be immensely helpful, however, some information and comparisons regarding the resulting data may be expanded to better highlight the utility. Additionally, all the data appears to all be collected from a single tissue of a rat brain with one replicate, which may somewhat misrepresent the reproducibility of the methodology. Below please find critiques which may be helpful to the authors. Overall, this workflow has immense potential.

We are grateful to the reviewer for the positive feedback and the helpful critiques raised by this thorough review. Further, we appreciate the recognition of our work as a valuable tool for the MSI community and believe its only the first step in many developments. Addressing the main concerns also of the other reviewers, we expanded our analysis to 4 replicates of rat brain cerebellum, to demonstrate the reproducibility of the SIMSEF methodology. Further, we adapted the acquisition method, to improve coverage of smaller metabolites. We have revised our manuscript accordingly and addressed the specific comments below.

Major

1. Results section, line 104 Can the authors estimate how long the time would be for processing the MS1 dataset to acquisition of the MS2 dataset? Often for LC-MS/MS this processing can be lengthy, and additionally the combination of MS1 and IMS across the MSI dataset tends to yield vary large datasets, so some considerations for sample and matrix stability within the source may need to be addressed since it is unclear how long it would take to process the data prior to the MS2 acquisition.

This assessment was indeed missing from the manuscript. We added Table S1, which contains summary statics on the number of extracted features and acquired MS2 spectra accompanied by time measurements for the individual steps. The order of the table rows reflects the order in the SIMSEF workflow.

Briefly, the MS1 data acquisition took 39-53 minutes. The processing time for MS1 data was between 9 and 14 min, while the SIMSEF scheduling required roughly 1-2 min. The MS2 spectra were acquired at a rate of roughly 2000 MS2 pixels per hour with multiple MS2 spectra per pixel. Resulting in a total acquisition time of about 3 - 3.5 h for 6000 - 7000 MS2 spectra. Here, 6 collision energies and 5 spectra per collision energy were scheduled per precursor. The MS2 acquisition time can be shortened substantially by more rigorous pre-filtering of the feature table, using fewer collision energies, or fewer MS2 spectra per energy.

Table S 1 Overview of time and feature statistics of four SIMSEF measurements on parallel tissue thin sections.

	R2c_1a	R2c_1b	R2c_1c	R2c_1d
MS1 pixels	16,161	16,765	13,329	17,984
Acquisition time / min	48.3	50.1	39.9	53.7
Feature detection time / min	9.3	12.0	9.6	14.3
MS1 features	1514	1652	1722	1769
SIMSEF scheduling time / min	1.0	1.8	1.2	2.3
MS2 acquisition time / min	186.0	206.0	205.0	221.0
MS2 pixels	6477	6835	6722	7192
Acquired MS2 pixels per hour / 1/h	2089.4	1990.8	1967.4	1952.6
Individual MS2 spectra per hour / 1/h	9327.7	8667.7	9180.3	8733.9

Since these times are multiple times lower than the stability of NEDC, which has been reported to produce stable signals for up to three days under vacuum (<https://pubs.acs.org/doi/full/10.1021/ac504294s>), there should be no issue concerning decreasing signal intensities.

2. Regarding Figure 2, it is unclear where this is meant to direct the reader with the choices in the figure legends, the pink and blue boxes are easy to identify but not the grey.

We harmonised the colours of the precursor isolation windows all to pale blue boxes now. Bold borders highlight the selected MS2 spectrum **d**, which is visualised below. We hope this makes it easier to distinguish.

For some of the lower IM values with the precursor isolation window, it almost appears that no signal is present.

This can be caused by the relatively high collision energy in that specific MS2 pixel. While 50 eV produced good MS2 spectra for the higher m/z, it might be too high for lower m/z ratios. It is expected that not every SIMSEF MS2 spectrum is informative, as it is almost impossible to predict the perfect collision energy for every precursor in untargeted analysis. Hence, the user can specify multiple CEs to increase the chance of at least one informative MS2 spectrum.

Furthermore, downstream processing in MZmine merges MS2 spectra that fall within the same mobility and m/z ranges. First for each collision energy separately and then all in one. The user can access both the raw spectra and multiple merged spectra variants. In general, the merging increases the spectral quality and even low intensity spectra play a role in this due to redundancy in signal detection.

We added this to the main text:

The results of one acquired TIMS ramp with 11 precursor ions are compiled in Figure 2, depicting the extraction of TIMS-resolved precursor-specific MS² spectra. Rule-based lipid annotation in MZmine annotated phosphatidic acid (PA) 18:0_18:1 based on lipid headgroup and fatty acyl (FA) chain fragments.^{39,40} Here, it should be noted, it is not expected that every TIMS-MS² pixel contains high-quality spectra for every precursor. The applied collision energy may be too high or too low for some molecular species, hence SIMSEF allows scheduling of multiple collision energies, to address this issue.

Would it be possible to add what the pixel looked like for the ion image this would have been generated from to better contextualize the resulting figure?

We agree it is hard to visualise these multi-dimensional datasets that combine ion mobility, m/z, and intensity. With the acquisition of the new dataset, we added the MS1 frame spectrum to the figure that collapses the whole ion mobility dimension of the corresponding MS1 frame into a single spectrum. This may aid in illustrating the spectral complexity that led to the selection of precursors. Adding the full MS1 heatmap would not show much additional information and would make the figure too large. The figure now nicely illustrates that SIMSEF is not just a common topN-DDA approach, since it does not just schedule the most intense signals, but also low-intensity features. Moreover, the selection takes the precursor m/z purity within the feature's ion mobility into account as well as the spatial distribution.

Figure 2: SIMSEF scheduled 11 precursor ions (magenta) for **a**, a TIMS-MS¹ frame spectrum, here visualized as accumulated spectrum with merged mobility dimension. The corresponding acquired MS² frame visualized as **b**, the total ion mobiligram showing the intensity distribution along the mobility dimension and **c**, the mobility-*m/z* heatmap of the TIMS-MS² pixel, showing precursor *m/z* isolation windows (magenta) and ion mobility MS² extraction windows for individual precursors (blue). **d**, an extracted TIMS-MS² scan for the precursor *m/z* 701.5121 within 1.29 - 1.33 Vs/cm² (bold blue) was annotated with the rule-based lipid annotation module and enriched by manual annotation of fragment signals. The observed headgroup fragment (*m/z* 152.9964) and signals for chain fragment ions (FA18:0; FA18:1) allow annotation of this precursor as PA 18:0_18:1. The applied collision energy produced high-quality fragment spectra for lipid species between *m/z* 400 – 850, while being too low for larger and too high for smaller molecules.

Additionally the SMART data reporting would be excellent to include along with any images displayed in the main text and the SI figures for transparency of this method: <https://pubmed.ncbi.nlm.nih.gov/36740651/>

This is a great suggestion to improve the readability of our results. We added the smart notation to the caption of the updated figure 3 and Supplementary Figure S4, S7, S12.

Figure 3: Composition of data and result views in the MZmine GUI. *a*, *b*, and *c* show spectral library matches to the public MoNA LC-MS/MS negative mode library. Spectral match to a reference spectrum of the metabolite glutathione (*a*), and the phosphatidyl serines PS 18:0_18:1 and PS 18:0_22:6. Multiple MS2 spectra were acquired across the tissue by MALDI-TIMS-SIMSEF. *d*, the feature table summarises all features by their properties (e.g., m/z and CCS) and adds interactive charts linked to the underlying data structures. SMART notation for images: Step size: 50 μm , spot size: 30 μm , identification confidence: MSI level 2, resolution: 40 000 FWHM @ m/z 1221, Time 48 min (R2c_1a), 50 min (R2c_1b), 40 min (R2c_1c), 54 min (R2c_1d)

3. Page 6, line 195, the presented range of CCS values seems vague given that the position of the double bond is not known, is there a chance this is multiple isomers? That may be important to clearly delineate a continued short coming to identification of some compound classes.

The CCS values reported in literature do not specify the double bond position or stereo chemistry, as the lipid is just specified as PS 36:1 and PS 40:6. The lipid nomenclature (see https://www.lipidmaps.org/shorthand_nomenclature/glycerophospholipids/pl_lpl, <https://doi.org/10.1194/jlr.S120001025>) clearly defines the notation for each identification level. As reported here, the double bond positioning remained ambiguous, therefore it is not reported.

The values of our example fall within a 3% error range to reported values that may be measured on other instruments.

Resolving double bond isomers by TIMS requires tuning of the instrument, sometimes for compound classes or even for pairs of compounds, to achieve the best separation. The TIMS ramp time could be increased or the ion mobility range could be decreased.

In the results section, we added more examples of lipids and metabolites and aim to describe the ambiguity better by pointing to the standard notation of lipids:

Measured CCS values of glutathione (162 Å²), PS 18:0_18:1 (286 Å²) and PS 18:0_22:6 (292 Å²) are within 3% error range with reported literature values for other IMS analysers (glutathione: 164.24 Å², PS 36:1: 289.4 Å², 279.1 Å²; PS 40:6: 295.4 Å², 284.9 Å²)^{41,42} It should be noted that the fatty acid chain composition was not specified in literature, which, including the different IMS setup, could explain the CCS discrepancies.

4. Regarding Figure S5, Can the authors estimate how many MS2 spectra yield good quality based on the data presented, the number of arrows seems low? Alternatively, how many features are in the MS1 dataset? It would be nice to make comparisons to MS 1 features, and then features that yield meaningful MS/MS spectra, followed by levels of annotation. This might provide the more general reader a greater grasp of the complexity of the MSI datasets.

We fully agree with the reviewer that the MS2 quality and the number of MS1 features are crucial points. We added a Supplementary Table to summarise various statistics on the number of TIMS-MS1 features and MS2 spectra. We agree that it is important to clearly demonstrate the quality of MS2 spectra. In our view this is not trivial but we applied multiple criteria to estimate the informational content and intensity contributions by multiple signals. Most spectra were found to be suitable for spectral library matching by having at least 4 signals with a base peak intensity > 10 signal-to-noise. Next, the BPI/TIC < 0.5 filters out spectra that only contain one major signal, contributing most of the intensity. We previously described quality metrics in the results section of the manuscript. With the replicate measurements, we created another supplementary table, regarding these metrics. Supplementary Table is a summary of the number of "quality" MS2 spectra and TIMS-MS imaging features extracted from all four datasets:

Table S 2: Summary of calculated quality criteria across four replicate SIMSEF experiments. BPI describes the base peak intensity.

	R2c_1a	R2c_1b	R2c_1c	R2c_1d
Features – total	1,514	1,652	1,722	1,769
Features – scheduled (matching SIMSEF criteria)	1,396	1,514	1,560	1,640
MS² – total	28,916	29,759	31,366	32,170
MS² – non-empty scans	23,444	24,813	26,076	25,705
MS² – at least 4 signals and BPI 10x S/N	17,887	19,126	19,537	19,599
"good" MS² – all the above and BPI / TIC < 0.5	12,540	13,481	13,555	13,867
Features – at least one non-empty MS²	1,329	1,426	1,556	1,460
Features – at least one "good" MS²	1,105	1,219	1,262	1,236
Features – "good" MS² / Features with non-empty MS²	83%	85%	81%	85%
Spectral library matches	100	121	118	101
Lipid annotation	192	223	208	212
Annotated features	208	255	238	232
Annotation rate	14%	15%	14%	13%

Furthermore, we added a histogram of the summed intensities within MS2 spectra.

Figure S: Histogram of the TIC distribution of all MS² spectra. Signals below the noise level of 100 were removed from MS2. An intensity of 1,000 (10x noise level) was used as a quality criterion for the base peak intensity in Supplementary Table S2.

5. It is slightly concerning that all this analysis appears across an N=1 experiment. It might be helpful in this evaluation to include consistency across multiple samples to better gauge effectiveness across different experiments.

We acknowledge that this was a short coming of the initial manuscript. Now we acquired four datasets and updated the manuscript accordingly. This includes supplementary tables that summarise the whole workflow and how much time each step took and the resulting number of features and MS2. Example screenshots compose three features in a feature table with images showing similar distribution patterns across all four samples linked to their spectral library match for annotation and matching rule-based lipid annotations. (see Figure 3 below)

Figure 3: Composition of data and result views in the MZmine GUI. **a**, **b**, and **c** show spectral library matches to the public MoNA LC-MS/MS negative mode library. Spectral match to a reference spectrum of the metabolite glutathione (**a**), and the phosphatidyl serines PS 18:0_18:1 (**b**) and PS 18:0_22:6 (**c**). Multiple MS2 spectra were acquired across the tissue by MALDI-TIMS-SIMSEF. **d**, the feature table summarises all features by their properties (e.g., m/z and CCS) and adds interactive charts linked to the underlying data structures. SMART notation for images: Step size: 50 μm , spot size: 30 μm , identification confidence: MSI level 2, resolution: 40 000 FWHM @m/z 1221, Time 48 min (R2c_1a), 50 min (R2c_1b), 40 min (R2c_1c), 54 min (R2c_1d).

Minor

1. Page 2, lines 40-42 This feels like the authors specifically mean to refer to MALDI or LDI based imaging approaches, for DESI/nanoDESI, LAESI, IRMALDESI this is simply not the case. MS/MS can be, and is collected as a part of these workflows.

Thanks for pointing out this difference between the technologies. In MALDI and LDI-based approaches, there are usually no possibilities to set up dataset-dependent MS2 acquisition. Some studies apply the DDA approaches that are designed for LC-MS2 to their imaging experiments. In our experience, this works but feels more like bending a system to something it was not designed for. This is why we developed the SIMSEF tools to make use of spatial and ion mobility information for each precursor ion.

Are there designated DDA methods for imaging DESI and the other techniques other than top-N approaches that you can recommend?

We tried to tone down our statement specifically referring to **MS1-only** studies and that **many** MS imaging studies **often** lack MS2 data for annotation. We also refer to the option to apply the DDA methods initially designed for LC-MS2.

A major bottleneck in many MS imaging studies remains the compound annotation, which often relies on accurate mass only, due to missing fragmentation data (MS²).^{2,6,7} Therefore, spectral library matching, manual annotation of fragmentation spectra, molecular networking,^{8,9} and other fragment ion-based approaches, such as molecular structure prediction in the SIRIUS software, remain unavailable for these MS¹-only workflows.¹⁰⁻¹² Generally, most public MS imaging studies lack MS² data because of the limited availability of spatially resolved data-dependent acquisition (DDA) modes, where MS² scans are usually scheduled based on the most abundant signals detected in MS¹.

2. Results section, line 108-109 This is more of a comment, depending on the desired spatial resolution, a simple offset of the laser is likely to result in the same results without any over or under sampling.

We agree that this will be the case. Actually, there are two modes of SIMSEF (See Figure S14). One divides each spot in four experiments with offset, and the other uses a smaller MS1 and a bigger MS2 spot. We chose the second approach because it would reserve a bigger area for the MS2 laser spot to ionise more material and increase the chance for meaningful MS2 spectra for low-intensity compounds.

3. Page 4, line 128 “lowest to highest area” This might be better qualified as peak intensity given that this could be mistaken for MSI area*

This is a good point. ‘Area’ was ambiguous and we now explain it more as the signal area (name in MZmine):

Initially, the MZmine feature table is sorted from the lowest to the highest signal area, i.e., the sum of all signal intensities across the sample.

4. Is Figure S8 the summary of all the features or just the lipid based subset? This seems very low compared to what could be achieved via LC-MS/MS of the extracted tissue. Again this speaks to how many MS/MS pixels from an imaging run may yield actionable information.

Indeed, this was only a subset of molecular networks to demonstrate that the dataset contained similar MS2 spectra and that we are now able to run MALDI-TIMS-MS2 molecular networking. We recreated the molecular networks with the new dataset (4 samples) and now show all subnetworks with at least 3 nodes. In the feature-based molecular networking step, we disabled the maximum cluster size (max nodes per network), to show the complexity of the spectral similarities in this lipid-based subset. Furthermore, we zoom into metabolite-based networks, to demonstrate the capabilities of SIMSEF for small molecule annotation in imaging experiments. In general, the discoverable chemical space by MALDI-MS imaging experiments can be shifted and expanded by using other matrices and maybe MALDI2 as post-ionisation.

Figure S 5: Feature-based molecular networks (FBMN) created from the SIMSEF imaging experiment using the GNPS ecosystem (minimum matched signals 4; minimum cosine similarity 0.7, maximum cluster size: off). These networks prove the gained data and annotation quality without the need to acquire additional LC-IMS-MS² data. Nodes describe ion image features with MS² spectra. A single representative, i.e., the most abundant MS² spectrum, was exported for each feature. Edges connect MS² spectra based on their modified cosine similarity (0.7-1.0), which is reflected by the edge weight. Diamond-shaped nodes are annotated by the MZmine rule-based lipid annotation module. Nodes with orange borders are annotated by spectral library matching to the MoNA, MassBank EU, MSDIAL Lipid Blast, or GNPS public libraries. Network a shows the lipid sub-network with many connections between annotated lipid species with a single fatty acyl chain substitution. Network b shows a small sub-network of annotated metabolites, such as nucleic acid diphosphates and phytic acid. The molecular networking job is available at <https://gnps.ucsd.edu/ProteoSAFe/status.jsp?task=9a6f6b34367f4ff69d81c9efe6aedd03>.

5. Page 10, line 301 I assume this is negative mode based on the matrix selection, but it should be expressly stated.

Thank you for pointing this out. We added this now to the Online methods in the MS1 image acquisition section:

MALDI-IMS-MS negative ion mode data were acquired on a[...]

Reviewer #3 (Remarks to the Author):

This manuscript, entitled "On-tissue dataset-dependent MALDI-TIMS-MS2 bioimaging", aims to develop a type of DDA for imaging mass spectrometry. The objective is to collect MS/MS in a more strategic fashion, thereby enhancing the number of ions that can be identified in an experiment. The authors first collect MS1 data, essentially at every other spot of the tissue. They then process the data using a custom script they wrote (integrated with MZmine 3). The software selects which ions should be fragmented where. They then acquire a second round of imaging data using MS2 on the spots of tissue remaining.

We thank the reviewer for their commitment and feedback. We value your interest in our work and the suggestions for improvement. The comments inspired new developments now accessible to MZmine and SIMSEF users. Furthermore, the whole evaluation was reworked with a new sample set of 4 parallel thin sections. We have revised our manuscript and addressed the specific points below.

I have two major questions related to the underlying premise put forward by the authors.

1. The authors argue that the number of compounds identified in timsTOF imaging is limited by a paucity of DDA modes. At least in our experiences, that has not really been the case. For us, the biggest limitation to identifying compounds is MS/MS sensitivity (especially when the tims is on) and an inability to mass resolve different molecular species. Simply put, no matter how the data is collected, the MS/MS is not high quality enough to substantiate high confidence identifications in most cases. I do not know that DDA helps that issue.

We agree with your comment that the sensitivity and quality of the acquired MS2 data is highly important. In our experience, the timsTOF-MS instrument is challenging to tune and has many parameters that need to be adjusted for specific compound classes. Especially the TIMS-on mode, offers great opportunities to resolve some of the interferences not resolved by the mass analyser.

Additionally, we have seen great improvements of the spectral quality after acquiring replicate MS2 spectra with multiple fragmentation energies. The downstream processing in MZmine merges MS2 spectra that fall within the same mobility and m/z ranges. First for each collision energy separately and then all in one. The user can access both the raw spectra and multiple merged spectra variants. In general, the merging increases the spectral quality and even low intensity spectra play a role in this due to redundancy in signal detection. Then, multiple high-quality MS2 that can be used for compound annotation. For the four new datasets, each precursor was fragmented at 6 discrete energies with 5 replicate MS2 each. We added Supplementary Table S1 and S2 to summarise the time needed to conduct the full acquisition workflows and the resulting number of features and MS2 spectra.

Currently, the alternative to our spatially aware dataset-dependent acquisition mode SIMSEF would be to acquire spectra manually or by broad band fragmentation. We have been using these alternative methods in various projects, but colleagues are moving new data acquisition to the SIMSEF workflow. One example project (not discussed in this manuscript) is the analysis of tattoo pigments in human skin samples by LDI-TIMS-MS2. Here, SIMSEF greatly reduced the analysis complexity described in this paper that uses broad band fragmentation (<https://doi.org/10.1016/j.aca.2023.340796>).

Regarding mass resolution, the chimeric MS2 spectra issue remains even in liquid chromatography-MS but is undoubtedly worse in MS imaging. We see great potential that IMS as an additional separation dimension in IMS-MS imaging can separate some isobaric interferences, but in the end, we cannot avoid detecting molecules with similar m/z or even the same molecular formula in the same mass scans. In our view, this

supports the need for sophisticated spatially resolved DDA in IMS-MS imaging studies, which was one of the key motivations for the development of SIMSEF. Chimeric MS2 of multiple lipids or other metabolites with the same molecular formula may be the only option for resolving those compounds if all other separation methods fail or are not an option. One of our examples in Supplementary Figure S9 demonstrates how different lipid chain fragments are detected in one MS2 scan of a feature that was identified as PS 38:1, leading to putative annotations for PS 20:1_18:0 and PS 20:0_18:1. This information would remain unavailable for TIMS-MS1 only imaging.

Figure S 9: SIMSEF data allows recognition of chimeric precursor features by annotating multiple compounds and lipid species in the same MS² scans. Spatial distribution of m/z 816.5726 \pm 0.01, mobilogram of the precursor feature, and two MS² spectra, which were annotated as PS 38:1. **a**, ion image of m/z 816.5726 \pm 0.01 in frontal sections of rat cerebrum, prepared analogous to the main manuscript. Green arrows indicate the spots of two MS² spectra acquired with a collision energy of -50 eV (**c**, **d**). **b**, the IMS-MS¹ imaging mobilogram of the precursor ion shows no indication for the presence of multiple isomers or other interferences. **c** and **d**, chimeric MS² spectra acquired in spots **c** and **d** contain fragment patterns of multiple lipid species. The observed fragment ions imply that multiple isomeric lipids with varying FA composition are present and overlap in their spatial distribution and ion mobility. Replicate MS² at different locations may indicate varying compositions of the interfering compounds. The user-defined list of fragmentation energies used by SIMSEF may guide follow-up studies investigating this precursor by full MS² image acquisition to better differentiate between the inhomogeneous distribution of the interfering ion species.

Just because one collects MS/MS on an ion, doesn't mean that it is above noise enough to merit a confident identification. If the MS/MS data cannot lead to a confident identification, then is it worth collecting? How many of the ~1100 ions for which MS/MS was collected led to confident identifications (the authors say they "annotated" 104 features but that term is often used to indicate low confidence assignments, which we can already get from MS1 data alone)? The MS/MS data should be shown and the identities should be included in some sort of table, together with MSI confidence scores. Given that the main argument is that DDA imaging leads to more identifications, it would be helpful to show tables of identifications with and without the DDA approach (together with confidence scores).

We agree that the provided annotations need more supporting material and rankings based on annotation levels defined by the Metabolomics Standards Initiative (MSI, <https://doi.org/10.1186/2047-217X-2-13>, <https://github.com/MSI-Metabolomics-Standards-Initiative/CIMR>). We would like to highlight that the MSI considers "annotated" the correct term for level 2, whereas "identified" level 1 requires authentic standards for every molecular species. Throughout the manuscript we made sure to use "*annotation*" when describing spectral library matches and rule-based lipid annotations. Most of our annotations were labelled as Level 2: Putatively annotated structure. Readers should still be aware that those annotations may be false considering the low coverage of reference compounds and their isomers in public spectral libraries. To be more precise regarding lipid annotations, we are using the standardised lipid short notation (see https://www.lipidmaps.org/shorthand_nomenclature/glycerophospholipids/pl_lpl, <https://doi.org/10.1194/jlr.S120001025>).

For better transparency, we added the SMART notation with MSI annotation level information to Figure 3:

Figure 3: Composition of data and result views in the MZmine GUI. a, b, and c show spectral library matches to the public MoNA LC-MS/MS negative mode library. Spectral match to a reference spectrum of the metabolite glutathione (a), and the phosphatidyl serines PS 18:0_18:1 and PS 18:0_22:6. Multiple MS2 spectra were acquired across the tissue by MALDI-TIMS-SIMSEF. d, the feature table summarises all features by their properties (e.g., m/z and CCS) and adds interactive charts linked to the underlying data structures. SMART notation for images[SH1]: Step size: 50 μm , spot size: 30 μm , identification confidence: MSI level 2, resolution: 40 000 FWHM @m/z 1221, Time 48 min (R2c_1a), 50 min (R2c_1b), 40 min (R2c_1c), 54 min (R2c_1d)

Annotations based on MS1 only, i.e. accurate m/z, isotope pattern, and adduct, is a level 4 annotation that can help in predicting the molecular formula of an ion. For structure annotation, however, additional information such as MS2 is needed and considered a standard approach.

This comment got us interested in how to generate annotation reports right in MZmine and we implemented a new module that exports all spectral library matches, images, and ion mobilogram plots to pixel and vector graphics. Furthermore, all annotations can be exported to a tabular format. The results are now found in the supporting file *features_summary.zip* and a few examples were also added as Figure 3 and Supplementary Figure S. Generally, MZmine's interactive user interface provides great options to manually validate annotations for significant features.

We added Supplementary Table S2 to summarise the number of MS2 spectra that fulfil multiple quality criteria and to provide annotation rates across all four samples. Each annotation is accompanied by supplementary images of the distribution, spectral library matches, and ion mobilograms.

Table S 2: Summary of calculated quality criteria across four replicate SIMSEF experiments.

	R2c_1a	R2c_1b	R2c_1c	R2c_1d
Features – total	1514	1652	1722	1769
Features – scheduled (matching SIMSEF criteria)	1396	1514	1560	1640
MS² – total	28,916	29,759	31,366	32,170
MS² – non-empty scans	23,444	24,813	26,076	25,705
MS² – at least 4 signals and BPI 10x S/N	17,887	19,126	19,537	19,599
“good” MS² – all the above and BPI / TIC < 0.5	12,540	13,481	13,555	13,867
Features – at least one non-empty MS2	1329	1426	1556	1460
Features – at least one "good" MS2	1105	1219	1262	1236
Features – “good” MS2 / Features with non-empty MS2	83%	85%	81%	85%
Spectral library matches	100	121	118	101
Lipid annotation	192	223	208	212
Annotated features	208	255	238	232
Annotation rate	14%	15%	14%	13%

2. The underlying premise of DDA is that you only have to collect MS/MS data on an ion one time. After it is identified from that MS/MS spectrum, the ion can then be annotated in other mass spectra. In LC/MS, the assumption is that an ion with a given m/z value across a chromatographic peak represents the same molecule. Or, in DDA, that an ion with the same accurate mass and retention time in different samples is the same molecule. While not always formally true, it is generally a good assumption (unless an ion elutes in the void volume). This is because the combination of LC retention times and accurate mass is specific. In imaging, however, the RT domain is lost. Now, one has to make conclusions about whether an ion at different locations of a tissue is the same or not based only on accurate mass. That assumption is precarious. Figure 9 c-d is a good example. The same ion is fragmented at different locations of the tissue, but the MS/MS patterns are clearly different.

This is a great summary of a general limitation of MS imaging and its dependence on accurate m/z to align signals across the whole sample. In our setup, the additional ion mobility dimension can improve but not solve this issue. Currently, integrative LC-IMS-MS² and IMS-MS² imaging analysis might provide the most comprehensive readout from a sample but with higher analytical complexity and time investment (<https://www.nature.com/articles/s41587-023-01690-2>). Using the LC data for alignment across samples, retention time, and statistical analysis considering the reduced matrix effects and the MS imaging data for localization of metabolites. Here, SIMSEF may help provide additional MS² data for alignment across the LC and imaging datasets. Generally, this integrative analysis workflow requires more than double the lab work and time to complete and can be added later in a study to confirm theories derived from a first SIMSEF IMS-MS² imaging study.

Regarding the premise that DDA captures single ions. We know from our experience with complex dissolved organic matter (DOM) samples and lipidomics that chimeric MS² spectra are common even in LC-MS² analysis. Therefore, it is not always possible to group MS² spectra to just one LC-MS feature and

be sure that features in another sample correspond to the same analyte. While alignment by MS2 similarity scoring is possible in MZmine, its application is usually limited by the low coverage of MS2 across samples and for low abundant features.

Our conclusion of Figure 9c-d is that now that we have MS2 data, we can annotate chimeric MS2 spectra and flag images most likely derived from multiple ions. The consequence would be to state that the depicted distribution is a sum of at least 2 compounds. A follow-up MS2 image with static precursor selection may use the best fragmentation energy determined by SIMSEF analysis. This remains impossible from MS1-only images.

We updated Figure 9 to better describe the conclusion and possible follow-up analyses:

Figure S 9: SIMSEF data allows recognition of chimeric precursor features by annotating multiple compounds and lipid species in the same MS² scans. Spatial distribution of m/z 816.5726 ± 0.01, mobilogram of the precursor feature, and two MS² spectra, which were annotated as PS 38:1. a, ion image of m/z 816.5726 ± 0.01 in frontal sections of rat cerebrum, prepared analogous to the main manuscript. Green arrows indicate the spots of two MS² spectra acquired with a collision energy of -50 eV (c, d). b, the IMS-MS¹ imaging mobilogram of the precursor ion shows no indication for the presence of multiple isomers or other interferences. c and d, chimeric MS² spectra acquired in spots c and d contain fragment patterns of multiple lipid species. The observed fragment ions imply that multiple isomeric lipids with varying FA composition are present and overlap in their spatial distribution and ion mobility. Replicate MS² at different locations may indicate varying compositions of the interfering compounds. The user-defined list of fragmentation energies used by SIMSEF may guide follow-up studies investigating this precursor by full MS² image acquisition to better differentiate between the inhomogeneous distribution of the interfering ion species.

So how does one know when the accurate mass is likely to be the same compound or a different compound? The authors suggest using pixel distances, but I have reservations about this logic. Imagine that an ion with an arbitrary accurate mass is detected at two different locations. In theory, MS/MS could be collected at one spot for identification. One could then hypothesize that ions at the second spot have the same identity. But how do we know that is true if we do not collect MS/MS? I'm not sure distance is a good proxy for assessing the probability of such an event. In heterogeneous tissues, a few micron shift could be sufficient to survey a different cell type. I actually think it's more likely that ions from distant cells of the same type are more likely to have the same identity than ions from neighboring cells of different cell types.

One way to potentially address this would be to collect MS/MS data for all pixels of an image. Then demonstrate that it looks the same within 20 pixels (or whatever distance threshold was used) but that it changes at further distances. It would be necessary to demonstrate this for multiple classes of compounds in different types of tissues. Without further support, the assumption that an m/z value within 20 pixels is the same molecule but that the same m/z value outside 20 pixels is not appears to be totally arbitrary.

We agree with the reviewer, that the absence of chromatography can lead to more chimeric spectra than in LC-MS experiments, and the same m/z could belong to different compounds in different pixels of the tissue/image. This is a general limitation that applies to all MS imaging studies. For this reason, we allow the acquisition of multiple MS² spectra per feature and collision energy. This sampling process is configurable and aims to acquire MS² from different regions. The minimum required distance (in pixels) between MS² scans of the same precursor m/z is configured by the user and can be decreased and the number of MS² per precursor can be increased to reach higher sampling rates. MZmine then annotates either all individual MS² or merged consensus MS² by spectral library matching and rule-based lipid annotations. These annotations can help to assess the purity of MALDI-TIMS-MS feature images.

Briefly, our method does not aim to solve the general issue of isobaric interferences in MS imaging nor do we think it is possible in a single experiment. Nevertheless, SIMSEF was intentionally developed to make use of all available data to try to avoid interferences, by assessing the purity of the quadrupole isolation window, using the ion mobility separation, and acquiring multiple MS² spectra per feature. Especially sampling of MS² from different regions of the sample is a great way to point the researcher to possible interferences and guide follow-up experiments like MS² imaging.

We developed a new module in MZmine to address this comment and provide more insight into potential chimeric MS² spectra and inhomogeneous distributions across the sample - discovered by comparing multiple MS² over the sampling area. We compare all MS² spectra of the same feature either intra-sample and/or across samples. This spectral similarity is calculated as the weighted cosine similarity of MS² spectra of the same collision energy, which were assigned to the same feature. This module filters out low-quality MS² spectra first. We plotted the results of these pair-wise MS² matching in the Supplementary Figure S10. Overall, 68 % of the acquired MS² spectra showed a cosine similarity of ≥ 0.7 with very few spectra below <0.4 . Additionally, MS² can be compared to spot differences in the mixture of isobaric interferences across different regions.

Additionally, a new MZmine module compares the similarity of MS² spectra of the same collision energy for every individual feature. This can help to identify chimeric distributions, if the fragmentation pattern changes within the MS² replicates across the tissue. In the present dataset, 68% of all MS² pairs had an intra-row similarity of ≥ 0.7 (see Figure S10).

Figure S10: Distribution of the cosine similarity of multiple MS² spectra assigned to the same TIMS-MS image feature in MZmine (Bin size = 0.02). Only MS² spectra of the same collision energy were scored against each other. Spectra were only scored if at least two spectra of the same collision energy were available. In total, 63,963 of 93,950 MS² pairs had a cosine similarity of ≥ 0.7 (68 %).

A few additional points:

3. The method for assessing chimeric MS/MS data is not complete. In many cases, two molecules will be similar enough in m/z that they cannot be mass resolved by the instrument even in MS1 mode (particularly when the resolution of the timstof is only $\sim 40k$). In this case, they will appear to be a single peak in MS1 data. MS/MS is needed to assess whether the composition of ions contributing to the peak is indeed the result of a single precursor. It's also possible that the peak is pure in the MS1 scan but then chimeric in the adjacent MS2 data, which is a few microns away. These possibilities are not taken into account.

This is a valid point. The instrument uses ion mobility separation to resolve isobaric or isomeric overlaps. However, not all isomers and isobaric interferences can be separated. Nevertheless, IMS offers an improvement compared to just MS imaging. Without resolving the precursor signal, there is no way to tell if a signal will result in a chimeric MS2 before acquiring it. The algorithm only tries to find sampling spots with the least possibility for contamination of the resulting spectra. Later, MS2 can be annotated to investigate the actual purity. In future versions, the purity scoring could be changed to evaluate the purity in all surrounding MS1 pixels, but this would also increase processing time.

We have added the above-mentioned module to calculate the intra-feature MS2 similarity within one sample or across samples to flag feature images that may contain inhomogeneously distributed ion species. This is directly visible in the feature table as the minimum, maximum, and average MS2 spectral cosine similarity and can be exported for all MS2 pairs.

Further we changed the main text in the SIMSEF - Precursor scheduling section to more clearly state the issue of co-isolation:

Furthermore, the algorithm considers criteria such as minimum intensity in a pixel, expected isolation purity, distance between MS² pixels, and ensures enough quadrupole m/z switching time. These filters aim to increase the chance for "pure" MS² spectra, but co-isolation of isobaric and isomeric interferences remains an issue that may be flagged and analysed during downstream processing in MZmine.

4. The MS1 data is processed off-line using a python script. For adjacent pixels, how much time lapses between the MS1 and MS2 experiments? I am wondering whether NEDC might undergo sublimation during that time, which would complicate the results considerably as it would change the molecular profile of the tissue.

This assessment was indeed missing from the manuscript. We added Table S1, which contains summary statistics on the number of extracted features, and acquired MS2 spectra accompanied by time measurements for the individual steps. The order of the table rows reflects the order in the SIMSEF workflow.

Briefly, the MS1 data acquisition took 39-53 minutes. The processing time for MS1 data was between 9 and 14 min, while the SIMSEF scheduling required roughly 1-2 min. The MS2 spectra were acquired at a rate of roughly 2000 MS2 pixels per hour with multiple MS2 spectra per pixel. Resulting in a total acquisition time of about 3 - 3.5 h for 6000 - 7000 MS2 spectra. Here, 6 collision energies and 5 spectra per collision energy were scheduled per precursor. The MS2 acquisition time can be shortened substantially by more rigorous pre-filtering of the feature table, using fewer collision energies, or fewer MS2 spectra per energy.

Table S 1 Overview of time and feature statistics of four SIMSEF measurements on parallel tissue thin sections.

	R2c_1a	R2c_1b	R2c_1c	R2c_1d
MS1 pixels	16,161	16,765	13,329	17,984
Acquisition time / min	48.3	50.1	39.9	53.7
Feature detection time / min	9.3	12.0	9.6	14.3
MS1 features	1514	1652	1722	1769
SIMSEF scheduling time / min	1.0	1.8	1.2	2.3
MS2 acquisition time / min	186.0	206.0	205.0	221.0
MS2 pixels	6477	6835	6722	7192
Acquired MS2 pixels per hour / 1/h	2089.4	1990.8	1967.4	1952.6
Individual MS2 spectra per hour / 1/h	9327.7	8667.7	9180.3	8733.9

Since these times are multiple times lower than the stability of NEDC, which has been reported to produce stable signals for up to three days under vacuum (<https://pubs.acs.org/doi/full/10.1021/ac504294s>), there should be no issue concerning decreasing signal intensities. The processing is indeed so fast that the samples can remain in the instrument. Timing and processing speeds were important considerations in developing this tool. The MS¹ data is processed in MZmine, which uses efficient memory handling and parallel processing.

5. The authors mention the use of a “prototype instrument control software”. Please clarify if this can be readily implemented by other people or what would be needed to use it on a Bruker tims systems as well as non-Bruker instruments.

Thanks for raising this point. We modified the *Code availability* section and documentation page to contain clear instructions on how to access all tools needed. All tools that we developed are open source and only the prototype instrument control software needs to be accessed through Bruker.

Briefly, the SIMSEF scheduling algorithm as shown in the preview, is completely open-source and implemented in MZmine. The SIMSEF task with all logic can be found here: https://github.com/mzmine/mzmine3/blob/master/src/main/java/io/github/mzmine/modules/tools/timstofmal_diacq/imaging/SimsefImagingSchedulerTask.java

The data acquisition Python script is open-source and available on GitHub https://github.com/SteffenHeu/simsef_py. A working binary with all dependencies is available for download there. Only the vendor library (Python wheel) had to be removed from the source code.

The timsControl 4.1 prototype is provided by Bruker Daltonics through the Bruker contact of the respective instrument owner. Previously, this was only described in the MZmine documentation page of the workflow. Now, we added this to the code availability section to clarify each part:

Code availability

MZmine modules are available on the official MZmine GitHub (<https://github.com/mzmine/mzmine3>) under the MIT open source licence. Documentation and a step-by-step guide are available in the official MZmine documentation (https://mzmine.github.io/mzmine_documentation/module_docs/tools_simsef/simsef.html, https://mzmine.github.io/mzmine_documentation/workflows/simsef/simsef_workflow.html).

The source code for the MS² acquisition tool is available on Zenodo ([DOI: 10.5281/zenodo.8009939](https://doi.org/10.5281/zenodo.8009939)) and GitHub (https://github.com/SteffenHeu/simsef_py). Working binaries are available on GitHub (https://github.com/SteffenHeu/simsef_py/releases). Bruker bindings were removed in the source code. To obtain the timsControl 4.1 prototype, researchers may contact their Bruker Daltonics representative.

Regarding other instruments, in theory, the MS₂ scheduling is independent of the TIMS-TOF-MS and may be applied to any dataset to produce lists of scheduled MS₂ spectra at XY coordinates with specific fragmentation energies. In practice, there are missing parts to enable SIMSEF for other instruments: Either direct control over the instrument data acquisition, like in our case for the Bruker timsTOF, or possibilities to read MS₂ schedules from files by other instrument control software. Generally, we are open to collaborations or contributions to our codebase. With experts and owners of other instrument types, our approach could be expanded in the future.

One reason why we chose the timsTOF, apart from availability, is the possibility of acquiring multiple MS₂ spectra in a single trapped-ion mobility ramp. So one pixel corresponds to multiple MS₂ spectra.

6. Why did the authors use NEDC as a matrix?

We used NEDC as a matrix because it showed good results with our setup according to prior method optimisation. We reached high ion yield with this matrix.

Minor comments:

Figure 2: Please clarify what “manually enriched” means and potentially add that the serine headgroup is observed as neutral loss.

Some fragment signals and neutral losses were automatically annotated, like the loss of the serine headgroup and the fatty acyl chain fragment signals. Other signals were annotated by hand to make the figure more informative. With the acquisition of the new dataset, we updated the figure to the new dataset. Here, the example is based on PA 18:0_18:1. We also clarified the figure caption.

Figure 2: SIMSEF scheduled 11 precursor ions (magenta) for **a**, a TIMS-MS¹ frame spectrum, here visualized as accumulated spectrum with merged mobility dimension. The corresponding acquired MS² frame visualized as **b**, the total ion mobilogram showing the intensity distribution along the mobility dimension and **c**, the mobility-*m/z* heatmap of the TIMS-MS² pixel, showing precursor *m/z* isolation windows (magenta) and ion mobility MS² extraction windows for individual precursors (blue). **d**, an extracted TIMS-MS² scan for the precursor *m/z* 701.5121 within 1.29 - 1.33 Vs/cm² (bold blue) was annotated with the rule-based lipid annotation module and enriched by manual annotation of fragment signals. The observed headgroup fragment (*m/z* 152.9964) and signals for chain fragment ions (FA18:0; FA18:1) allow annotation of this precursor as PA 18:0_18:1. The applied collision energy produced high-quality fragment spectra for lipid species between *m/z* 400 – 850, while being too low for larger and too high for smaller molecules.

Line 182: Does “consensus spectra” mean averaged spectra?

MZmine merges consensus spectra with the same collision energy and across energies by taking the maximum intensity of an *m/z* signal across all merged scans. This is done to retain the general amplitude and intensity scale for the resulting spectra.

We added clarification on how spectra are averaged to the main text:

The MS² scans are extracted for each pixel and then optionally merged into consensus spectra with the same collision energy and across energies taking the maximum intensity of a signal across all scans.

Line 238: should be “in Method 3. Method 4 decreases...”

Thank you for catching this error.

Line 306: Is the raster size for MS1 really 50 μm ? Given the description it seems very large, especially since later for MS2 the laser spot size was increased to 50 μm .

We added the SMART notation to images like Figure 3 to better describe the final resolution and laser spot sizes. The raster size corresponds to the stage movement and was indeed 50 μm . The laser size was 30 μm in MS1 and 50 μm in MS2 experiments, as detailed in the methods section:

Imaging data were acquired with a laser field size of 30 μm . The imaging raster size was 50 μm to keep material for subsequent MS² acquisition.

Since the image was acquired across the whole section, a 50 μm raster size is a good compromise between resolution and acquisition time. We performed experiments with smaller raster sizes of 20 μm (the technical limit of the instrument) and also achieved MS2 spectra of similar quality. This was acquired on a different tissue type and in positive rather than negative ion mode. We believe adding this would bloat the paper unnecessarily. We would keep the scope of this paper to the routine acquisition of MS2 data and not the acquisition of high-resolution images that have been demonstrated by others with sub-micron resolutions.

I recommend calling out supplemental figures in order (Figure S7 is currently not referenced in the main text).

Thank you for pointing this out. We now refer to all Supplementary Elements in their order.

Figure S1: Make sure that all options are logical. There are a few points in the flow diagram where the other option (Yes or No) is not accounted for. Some also seem to have an answer different from yes or no (eg, “Determine which collision energy has the fewest MS2 pixel for feature”).

We appreciate the thoroughness as this is an important part of understanding the flow of the algorithm. We revised Figure S1:

Figure S 1: Flow diagram of the SIMSEF algorithm.

Figure S1: typo in “Find existing MS2 pixels”

Thank you for the thorough read.

Figure S4: “Minimum chimerity score” seems confusing. Should it be “Maximum chimerity score”?

This was indeed unclear. Our initial name for this parameter was confusing and we, therefore, decided to rename it to “Minimum purity score”. The purity is calculated by dividing the main signal by the summed intensity of all signals within the precursor isolation width (another parameter). We hope this improves the clarity.

Reviewers' Comments:

Reviewer #1:

Remarks to the Author:

The authors have done an excellent job in revising their manuscript very thoroughly: Additional experiments have been performed to improve the (visual) quality. Refs have been revised. The text has been revised wherever it seemed to lack clarity. Additional and less trivial metabolites are now included in the MS/MS analysis.

Congratulations for a manuscript with a lot of upside.

Reviewer #2:

Remarks to the Author:

All my prior concerns have been addressed. I would like to commend the authors for expanding their analyses to 4 tissue sections and adding the additional tables on data acquisition times and MS/MS spectral quality assessment. This has truly demonstrated that the methodology is robust and capable of greatly expanding the utility of MSI to directly gather data for compound identification within a single experiment. I look forward to implementing this in my lab. In response to the authors question 'Are there designated DDA methods for imaging DESI and the other techniques other than top-N approaches that you can recommend?' In this case, the authors are correct in asserting that existing methods are based on LC-MS/MS DDA and only operate as a top-N for selection and do not take the spatial distribution of the signal into account. Apologies for the confusion there.

I would also like to comment that the authors have handled the identifications and annotations critiques raised by the other reviewers well, this paper and method does not seek to augment existing challenges with data annotation, rather its directly facilitating how we might gather this data at scale in a spatially informed manner, which is critical in being able to further how we handle this data to improve compound identification. I do not expect these authors to solve identification and annotation challenges in this single manuscript that have plagued the field for decades.

Reviewer #3:

Remarks to the Author:

I thank the authors for carefully considering each of my comments. They have done a thorough job of addressing each. All of my concerns have now been addressed, within the limitations of MSI. One potential further suggestion might be to include a paragraph in the Discussion on limitations, so that some of the ideas outlined in the response document could be further incorporated into the manuscript. However, I defer to the authors' judgement on what they think is best. While this manuscript doesn't solve every issue with MSI, it is a step forward and I support publication.

Reviewer #1 (Remarks to the Author):

The authors have done an excellent job in revising their manuscript very thoroughly: Additional experiments have been performed to improve the (visual) quality. Refs have been revised. The text has been revised wherever it seemed to lack clarity. Additional and less trivial metabolites are now included in the MS/MS analysis.

Congratulations for a manuscript with a lot of upside.

We want to thank the reviewer for their thorough comments that helped us improve the manuscript greatly.

Reviewer #2 (Remarks to the Author):

All my prior concerns have been addressed. I would like to commend the authors for expanding their analyses to 4 tissue sections and adding the additional tables on data acquisition times and MS/MS spectral quality assessment. This has truly demonstrated that the methodology is robust and capable of greatly expanding the utility of MSI to directly gather data for compound identification within a single experiment. I look forward to implementing this in my lab.

We are happy to assist in any way and to discuss further improvements to our workflow.

In response to the authors question 'Are there designated DDA methods for imaging DESI and the other techniques other than top-N approaches that you can recommend?' In this case, the authors are correct in asserting that existing methods are based on LC-MS/MS DDA and only operate as a top-N for selection and do not take the spatial distribution of the signal into account. Apologies for the confusion there.

Thanks for the clarifications. We thought maybe we missed a new tool or the configuration options on other instruments as this is a quite dynamic field. That was our impression that most tools just employ the LC-MS/MS DDA algorithms, which work well but leave space for technical improvements like those described in our paper.

I would also like to comment that the authors have handled the identifications and annotations critiques raised by the other reviewers well, this paper and method does not seek to augment existing challenges with data annotation, rather its directly facilitating how we might gather this data at scale in a spatially informed manner, which is critical in being able to further how we handle this data to improve compound identification. I do not expect these authors to solve identification and annotation challenges in this single manuscript that have plagued the field for decades.

Thank you for strengthening our point on this. Indeed, high quality MS2 spectra can only be a first step, maybe even a requirement to high confident compound annotation.

Reviewer #3 (Remarks to the Author):

I thank the authors for carefully considering each of my comments. They have done a thorough job of addressing each. All of my concerns have now been addressed, within the limitations of MSI. One potential further suggestion might be to include a paragraph in the Discussion on limitations, so that some of the ideas outlined in the response document could be further incorporated into the manuscript. However, I defer to the authors' judgement on what they think is best. While this manuscript doesn't solve every issue with MSI, it is a step forward and I support publication.

We want to thank the reviewer for their outstanding work and thorough comments that greatly impacted the quality of our final manuscript. Thank you for supporting us. We added a short sentence on the challenging separation of compounds that may limit annotation capabilities if chimeric spectra are acquired.